# ‘There and Back Again’—Forward Genetics and Reverse Phenotyping in Pulmonary Arterial Hypertension

**DOI:** 10.3390/genes11121408

**Published:** 2020-11-26

**Authors:** Emilia M. Swietlik, Matina Prapa, Jennifer M. Martin, Divya Pandya, Kathryn Auckland, Nicholas W. Morrell, Stefan Gräf

**Affiliations:** 1Department of Medicine, University of Cambridge, Cambridge Biomedical Campus, Cambridge CB2 0QQ, UK; es740@cam.ac.uk (E.M.S.); mp931@medschl.cam.ac.uk (M.P.); jm2055@medschl.cam.ac.uk (J.M.M.); dp623@medschl.cam.ac.uk (D.P.); ka328@cam.ac.uk (K.A.); nwm23@cam.ac.uk (N.W.M.); 2Royal Papworth Hospital NHS Foundation Trust, Cambridge CB2 0AY, UK; 3Addenbrooke’s Hospital NHS Foundation Trust, Cambridge CB2 0QQ, UK; 4NIHR BioResource for Translational Research, University of Cambridge, Cambridge Biomedical Campus, Cambridge CB2 0QQ, UK; 5Department of Haematology, University of Cambridge, Cambridge Biomedical Campus, Cambridge CB2 0PT, UK

**Keywords:** forward phenotyping, forward genetics, reverse genetics, reverse phenotyping, pulmonary arterial hypertension, intermediate phenotypes, whole-genome sequencing, epigenetic inheritance, genetic heterogeneity, phenotypic heterogeneity

## Abstract

Although the invention of right heart catheterisation in the 1950s enabled accurate clinical diagnosis of pulmonary arterial hypertension (PAH), it was not until 2000 when the landmark discovery of the causative role of bone morphogenetic protein receptor type II (*BMPR2*) mutations shed new light on the pathogenesis of PAH. Since then several genes have been discovered, which now account for around 25% of cases with the clinical diagnosis of idiopathic PAH. Despite the ongoing efforts, in the majority of patients the cause of the disease remains elusive, a phenomenon often referred to as “missing heritability”. In this review, we discuss research approaches to uncover the genetic architecture of PAH starting with forward phenotyping, which in a research setting should focus on stable intermediate phenotypes, forward and reverse genetics, and finally reverse phenotyping. We then discuss potential sources of “missing heritability” and how functional genomics and multi-omics methods are employed to tackle this problem.

## 1. Introduction

Rare diseases, such as pulmonary arterial hypertension (PAH), are enriched with underlying genetic causes and are defined as life-threatening or chronically debilitating disorders with a prevalence of less than 1 in 2000 [1]. Although individually characterised by low prevalence, in total, rare diseases pose a significant burden to health care systems and a diagnostic challenge. Over 6000 rare diseases have been reported to date (ORPHANET [2]) and new genotype-phenotype associations are discovered every month [3]. Despite the several-fold increase in genetic diagnoses in the area of rare diseases [4], the cause of the disease remains elusive in a significant proportion of cases.

Since the 2008 World Symposium on Pulmonary Hypertension (WSPH), the term “heritable PAH” (HPAH) has been used to describe both familial PAH and sporadic PAH with an identified underlying pathogenic variant [5] (Table 1). Family history is a vital disease component directly linked to the proportion of variation attributable to genetic factors, known as heritability [6,7]. Familial studies have been used historically as a tool for gene mapping, with the classical example of twin studies commonly used to disentangle the relative contribution of genes and environment to complex human traits [2].

In statistical terms, heritability is defined as a proportion of the phenotypic variance that can be attributed to the variance of genotypic values:(1)$$H^{2}\;\left(broad\;sense\right)\;=\;Var\;\left(G\right)\;÷\;Var\;\left(p\right)$$

*H*^2^ estimates are specific to the population, disease and circumstances on which they are estimated [8]. In theory, total genotypic variance (Var(G)) can be divided into multiple components: Var(A)—total additive variance (breeding values); Var(D) —dominance variance (interactions between alleles at the same locus); Var(I) —epistatic variance (interactions between alleles at different loci); Var(G×E)—variation arising from interactions between genes and the environment.
(2)Var (G) = Var (A) + Var (D) + Var (I) + Var (G × E)

In practice, total genotypic variation is difficult to measure, although estimates can be made, and the components of genotypic variation are nearly unattainable. Hence genetic studies usually refer to heritability in its narrow sense, which is the proportion of the phenotypic variance that can be attributed to favourable or unfavourable alleles.
(3)$$h^{2}\;\left(narrow\;sense\right)\;=\;Var\;\left(A\right)\;÷\;Var\;\left(p\right)$$

Recently, genome-wide association studies (GWAS) have enabled the estimation of additive heritability attributed to common genetic variation (single nucleotide polymorphisms—SNPs), albeit with a typically small effect size [9]. This has led to the issue of “missing heritability” [10], whereby SNP-based estimates were not sufficient to explain prior heritability predictions arising from twin or familial recurrence studies. To that effect, several hypotheses have been proposed (Figure 1 and Figure 2) including the complex interplay between genes and environment and the often overlooked potential contribution of structural variation [11].

To date, about 75% of patients with a clinical diagnosis of idiopathic pulmonary arterial hypertension (IPAH) have no defined genetic cause of the disease. This review outlines the role of forward and reverse genomic and phenomic approaches as well as other omic technologies in the search for missing heritability of PAH (Figure 1). Understanding the genetic architecture of PAH and its dynamic interplay with the environment is a prerequisite to predict personalized patient risk, decipher interaction pathways underlying the disease and develop strategies for therapy and prevention.

## 2. Genetic and Phenotypic Heterogeneity

Discoveries in the field of rare diseases have been hampered by genetic and phenotypic heterogeneity of these entities. Genetic heterogeneity is a situation in which sequence variation in two or more genes results in the same or very similar phenotype. The degree of genetic heterogeneity varies between different diseases (Figure 3A). For instance, sickle cell anaemia has only been associated with mutations in one gene [19], Haemoglobin subunit β (*HBB*) and tuberous sclerosis [20] in two genes, retinitis pigmentosa [21] with over 60 and intellectual disability with over 800 [22]. PAH also shows high genetic heterogeneity (Figure 3B), with so far around ~20 risk genes reported [23] and more expected to be found.

Genotypic heterogeneity is further complicated by phenotypic heterogeneity. Pulmonary hypertension (PH) is a highly heterogeneous condition, defined as an elevation of mean pulmonary artery pressure (mPAP) equal to or greater than 25 mmHg measured by right heart catheterisation (RHC) in the supine position at rest [17]. This somewhat arbitrary threshold for defining PH was proposed at the 1st WSPH in Geneva in 1973 and has now been challenged by a new body of evidence showing that normal mPAP is 14 ± 3.3 mmHg, which suggests an upper limit of normal at 20 mmHg (14 mmHg + 2SD). This new threshold has been endorsed by the 6th WSPH along with the inclusion of pulmonary vascular resistance (PVR) and pulmonary artery wedge pressure (PAWP) cut-offs into the new haemodynamic definition. This new definition categorises PH into three groups based on haemodynamic criteria: pre-capillary PH, isolated post-capillary PH (IpcPH) and combined pre- and post-capillary PH (CpcPH) [18] (Table 1).

Haemodynamic definitions of PH encompass multiple cardiopulmonary entities that have been classified into five groups: PAH, PH secondary to left heart disease, PH due to lung disease and/or hypoxia, chronic thromboembolic pulmonary hypertension (CTEPH) or other pulmonary artery obstructions, and PH with unclear and/or multifactorial mechanisms.

The clinical classification aims to categorise patients into groups according to pathophysiological mechanisms, haemodynamics and therapeutic management. Despite this, there is persistent heterogeneity within groups and subgroups, and not all patients fit easily into a single category, which can be a reflection of genetic pleiotropy (Figure 4).

Based on the recent advances in mechanistic understanding of the disease, the 6th WSPH proposed new changes to the classification of PH, including two previously recognised phenotypes as the subgroups of Group 1, namely “PAH long-term responders to calcium channel blockers” (Group 1.5) and “PAH with overt features of venous/capillaries (Pulmonary veno-occlusive disease(PVOD)/Pulmonary capillary haemangiomatosis (PCH)) involvement” (Group 1.6) [18]. Other phenotypes previously reported in the literature and summarised in An Official American Society Statement: Pulmonary Hypertension Phenotypes [27] such as “severe” PH in respiratory disease, maladaptive right ventricular (RV) hypertrophy, PH in elderly individuals, PAH in children and PAH with metabolic syndrome [28] are awaiting clinical validation and confirmation of utility both in clinical practice and research settings.

## 3. Forward Phenotyping for Genetic Studies

A precise definition of the phenotype of interest is a cornerstone of any genetic study (Figure 1, process 1). As described above, clinical diagnosis relies on clustering patients based on observable and measurable traits, signs and symptoms, which are the product of genetic, epigenetic and environmental factors. As a consequence, clinical phenotypes can be dynamic and reactive, which is useful and desirable in the clinical setting but unsuitable for genetic studies. A distinction must be made between clinical and research diagnosis, particularly diagnosis for genetic analysis. The former is usually spread over time and acquired in several stages: history taking, physical examination, differential diagnosis and confirmation, the latter usually needs to be ascertained during a single encounter. To make this feasible and reliable, standardised checklist and operating procedures need to be in place, diagnostic criteria should follow simple inclusion/exclusion rules and phenotypes need to be described using controlled vocabulary to avoid ambiguity. Additionally, the data must be in a format amenable to computational analysis. Finally, the validity of phenotypes is confirmed in test cohorts, through functional studies and ultimately via reverse phenotyping (see below). The accuracy and precision of phenotype measurements are of paramount importance. In genetic studies, the diagnosis misclassification or admixture of phenocopies can significantly affect power to detect an association [29]. Equally, categorising biologically continuous phenotypes (i.e., age, mPAP, diffusing capacity of the lungs for carbon monoxide (DLCO)) is prone to errors due to flaws in quantification methods and arbitrary thresholds.

Phenotype optimisation for genetic studies aims at finding homogenous groups of patients that likely share the same genetic architecture. This can be approached through various strategies. For example, an extreme phenotype strategy aims at identifying rare variants with large effect sizes through recruitment of patients with traits at either end of the phenotypic spectrum. These phenotypes can be based on family history, age of onset, outcome, severity scores, biomarker levels, disease trajectory or response to treatment [30,31,32]. Such stratification was proven to increase the power to detect novel disease risk genes [30,33,34] and to be cost-effective [35]. Other strategies include covariate-based methods which jointly estimate the effect of multiple variables and data reduction techniques. Alternatively, intermediate phenotypes can be used. Intermediate phenotypes are features closer to underlying biology that are at least as heritable as the phenotype itself, stable over time, and are associated with the disease of interest [36].

Although clinical phenotyping remains the most widely used method of patient stratification both in clinical practice and research, it requires substantial domain knowledge and is time-consuming. Computational phenotyping based on clinical and/or “omics” datasets using machine learning might be an alternative due to unparalleled diagnostic precision, accuracy and speed. Two recent publications exemplify the power of computational tools in identifying disease phenotypes. Based on blood cytokine profiles, a prospective observational study of Group 1 PAH discovered and validated four immune phenotypes; importantly, these phenotypes differed in clinical outcomes despite the fact that demographics, PAH aetiologies, comorbidities, and treatments were similar across clusters [37]. Likewise, clinical data mining using the Comparative Prospective Registry of Newly Initiated Therapies for Pulmonary Hypertension (COMPERA) again revealed four clusters with differing survival and response to therapy [38]. A viable alternative approach for clinical data mining is utilising phenotype ontologies, such as Human Phenotype Ontology (HPO), which allow standardised, highly granular and precise phenotyping across different disease domains [39]. Use of ontologies to define phenotypes has already proven useful in identifying novel candidate genes for rare disorders [40]. Ontology-based analysis of phenotypes has been further facilitated by the implementation of methods for manipulation, visualisation and computation of semantic similarity between ontological terms and sets of terms [41].

## 4. Forward Genetics

### 4.1. Concepts

Forward genetics is a classic molecular genetics approach used to elucidate the genetic underpinnings of a mutant phenotype of interest [42]. Forward genetics is typically considered a ‘phenotype to genotype’ approach as mutant phenotypes are first observed before their corresponding genes are identified (Figure 1, process 2). In humans, forward genetics approaches most commonly include family-based linkage studies and/or genome-wide association studies (GWAS) and, more recently, rare variant association studies (RVAS).

### 4.2. Methodology

#### 4.2.1. Study Design

The two main approaches for studying the underlying genetics of PAH are family-based studies and case-control studies. The former is based on studying inheritance patterns of genetic polymorphisms, the second involves comparing genotype frequencies between cases and controls. Family-based studies are effective when parental samples along with phenotype information are available and the disease in question has a high penetrance; they are particularly useful for studying dichotomous traits and are robust to population stratification [43]. Case-control studies are a viable alternative if the above criteria are not met, although they have their own challenges which need to be addressed. To name the most important:Selection of cases (recognition of selection bias, incident vs. prevalent cases recruitment)Case definition (precise definition of the phenotype that can be ascertained in a research setting)Selection of controls (healthy vs. disease controls, matched in respect to age, sex and ethnicity, having a comparable evaluation of presence or absence of the phenotype in question)

Power calculations in genetic studies are an absolute necessity as ignoring this basic step can lead to both underpowered (risking false rejection of null hypothesis and characterised by wide sampling distributions for sample estimates) and overpowered (wasteful and often unethical) experiments. Factors limiting power to detect new genotype-phenotype associations which need to be accounted for are phenotypic variance, phenocopies, the effect size of risk alleles and minor allele frequency (MAF), with the last two factors driving the difference of sample sizes between GWAS and RVAS.

#### 4.2.2. Statistical Methods

Prior to the widespread use of GWAS, the most important tool in genetics were linkage studies in families, these were particularly useful in single-gene disorders in which implicated genes have large effect sizes. GWAS on the other hand compares the frequency of common SNPs between unrelated cases and controls. The associated SNPs are then considered markers of relevant regions that influence the risk of the phenotype. In fact, power calculations provided evidence that GWAS are better than linkage studies at detecting variation with small effect sizes [44]. Multiple statistical methods can be applied in GWAS, for example, Pearson X^2^ test, normal approximation to Fisher’s Exact Test, logistic regression, categorical model tests, Cochran–Armitage Trend test, and allele tests. The best method depends on the mode of inheritance and trait frequency. Importantly, the assumptions used in various tests may differ; these assumptions directly impact the results, as tests that assume the same mode of inheritance should yield the same results (i.e., Cochran–Armitage Trend Additive test, Logistic Regression Additive test and Allele test). Due to a large number of comparisons, adjustment for multiple testing is necessary; therefore, the *p*-value threshold is Bonferroni corrected (which encourages high type II error). Additionally, genotype and phenotype misclassification errors impact on power in GWAS. Epistatic scenarios and modelling gene–environment interactions require yet another set of methods and are computationally challenging although feasible [45].

GWAS is unsuitable for single-variant testing, due to the potentially low prevalence of mutation carriers and small effect size, which would both require unfeasibly large sample sizes. Instead, gene and region-based aggregation approaches have been developed which compare mutation frequencies between cases and controls within the boundaries of the gene. These techniques are appropriate when different variants exhibit an equal risk of disease and thus have the same phenotypic impact. For instance, several variants may result in LoF (e.g., nonsense, frameshift, essential splice site), and thus analysis would determine the association by counting the presence of LoF variants between cases and controls. Prior to association testing, quality control and filtering methods are utilised, namely sequencing quality scores, MAF filters [46] (usually MAF of 1:10,000 for autosomal dominant disorders and MAF of 1:1000 for autosomal recessive disorders) and in silico predictions. Predictions include deleteriousness scores for missense variants such as PolyPhen-2 [47], Sorting Intolerant From Tolerant (SIFT) [47,48], and rare exome variant ensemble learner (REVEL) [49], conservation scores such as Genomic Evolutionary Rate Profiling (GERP) [50], PhyloP [51] or PhastCons [52], or the Combined Annotation Dependent Depletion (CADD) score [53], which combines several metrics in one score. Analysis of the protein-coding region, consisting of ~20,000 genes, requires adjustment for multiple-testing. This can be done using the Bonferroni correction, where α = (0.05/20,000) ≈ 2.5 × 10^−6^). Where several models are applied, this adjustment must be made more stringent by dividing by the number of models tested. Region-based collapsing approaches hinge on the notion that different regions within genes may vary in their tolerance to missense variation. An alternative approach, particularly useful in smaller studies, is collapsing variants that belong to the same gene set (i.e., genes that belong to the same pathway). Candidate gene testing is a powerful approach to avoid overcorrection and, therefore, false-negative results. This proved useful when investigating members of the transforming growth factor-β (TGF-β) pathway, such as *SMAD9* [54], which did not reach statistical significance in the exome-wide analysis [24]. More recently, the same approach revealed an association between *TET2* and PAH, which was further supported by experimental evidence [55] but did not reach exome-wide significance [25].

Complex genetic models such as recessive inheritance pose additional challenges. In the recessive mode of inheritance, the MAF threshold must be more lenient as heterozygotes are unaffected (higher MAF in reference populations); also, variants in cis configuration (affecting the same allele) might be wrongly counted (as a pose to trans variants, which are those present on opposing alleles). Similarly, testing for digenic inheritance is particularly problematic due to the large number of possible combinations requiring testing and adjusting for [56].

A number of statistical methods have been developed to test for rare variant associations. Burden tests [57,58,59] aggregate the information found within a predefined genetic region into a summary dose variable. In weighted burden tests [60], variants are weighted according to their frequency or functional significance. Adaptive burden tests [61] aim to account for bidirectional effects by selecting appropriate weights. Variance component (kernel) tests such as (Sequence) Kernel Association Test (SKAT) [62] allow to test risk and protective variants simultaneously but are underpowered when most variants are causal, and effects are unidirectional. Omnibus tests such as SKAT-O [63], which combines burden tests with the variance-component test, might be particularly useful when there is little knowledge of the underlying disease architecture. In addition to frequentist approaches, a Bayesian statistical framework offers a robust alternative. Bayesian model comparison methods such as BeviMed [64] allow for the testing of associations between rare Mendelian disease and a genomic locus by comparing support for a model where disease risks depend on genotypes at rare variant sites in the locus and a genotype-independent “null” model. The prior probability in such models can vary across variants (reflective of external biological information, i.e., depending on MAF, conservation scores, gene ontologies, expression in the tissue of interest) or be constant for all genes/variants reflecting the prior belief of the overall proportion of variants that are associated with a given phenotype.

Last but not least, an essential step in rare variant discovery is to ascertain the pathogenicity of a given variant and its causative role in the disease. Not all damaging variants are pathogenic and in silico approaches alone are not enough to predict if the variant is disease-causing [65]. Viability and phenotyping inferred from knockout mice screens, as well as essentiality screens on human cell lines, may further help predict variant impact [66]. To aid both research and clinical decision making, the American College of Medical Genetics and the Association for Molecular Pathology (ACMG) issued recommendations that combine and weigh the computational, functional, population and clinical evidence to determine pathogenicity [67]. Other initiatives such as ClinGen and ClinVar aim to define the clinical relevance of genes and variants reported in the literature for use in precision medicine and research [68].

#### 4.2.3. Molecular Genetic Techniques

Molecular genetics techniques used for genetic diagnosis, including the detection of specific gene mutations and copy number variants, have been recently summarized [69]. Traditional methods used to identify candidate genes involved in the pathogenesis of PAH include linkage analysis, but more recently next-generation sequencing (NGS) has taken centre stage. The advent of NGS technologies has opened a plethora of opportunities both for clinical diagnostics and research. Such technologies that fall under the umbrella of NGS include targeted panel sequencing, whole-exome sequencing (WES) and whole-genome sequencing (WGS). Nevertheless, the need for more conventional methods such as Sanger sequencing and multiplex ligand-dependent probe amplification (MLPA) remains. Disease-specific panels include a set of genes or regions of genes that are known to be causative of a specific phenotype. This is particularly beneficial in the clinical context when assessing highly heterogeneous traits, such as intellectual disability. Although these panels are not always consistent across laboratories, efforts are being made to produce guidance around their design and development [70]. Targeted panel testing has been introduced in PAH including known and candidate disease genes [71].

WES includes < 2% of the genome i.e., the coding regions only. This method is clinically useful given that 85% of all described disease-causing sequence variants are in this region [72]. For diseases that are more genetically heterogeneous, WES has proven to be a fruitful method, especially when incorporating segregation analysis, which increases the diagnostic yield from 23.6% in probands to 31% in child–parent trios [73]. WES has been used for both the identification and discovery of candidate genes in PAH and has been applied to family-based [74] and case-control studies [75]. Limitations of WES include poor coverage of some exons such as GC rich regions and low confidence to identify structural variation [76].

WGS is a high-throughput sequencing technology predominantly used in the research setting [77]. WGS is massively parallel, DNA fragments are aligned to form a contiguous sequence. The cost of this technology is halving approximately every two years [78]; however, the $1000 genome often referred to is still some way off unless sequencing is performed at scale [79]; as an example, WGS is currently being introduced to the UK National Health Service in collaboration with Genomics England.

European Respiratory Society and European Society of Cardiology (ERS/ESC) guidelines published in 2015 recommended sequential testing starting with bone morphogenetic protein receptor type-2 (*BMPR2*) sequencing and MLPA in patients with sporadic/familial PAH and *EIF2AK4* sequencing in sporadic/familial PVOD/PCH [17], and this approach has been successfully used in many clinical and research settings across the world [80]. With the decreasing cost of WES/WGS, a common practice is now that of virtual panel testing whereby a selected number of genes are chosen for bioinformatics analysis based on the individual’s phenotype. This also allows for data reanalysis when a novel disease-gene is identified and/or another condition is suspected (emerging phenotype over time). In the UK, this is coordinated at a national level via the Genomics England PanelApp tool; virtual disease gene panels applied to WGS data are continuously curated and include a PAH panel [81]. As more patients with PAH are being tested via NGS methods, a diagnostic benefit is starting to emerge (Figure 3C).

Despite the benefits of NGS technologies, there are some challenges that require attention and systematic solutions, among these, storage and handling of big data remains a significant consideration, also management of incidental findings as well as the reporting of variants of unknown significance (VUS).

Along with the growing number of research projects using NGS to uncover the genetic basis of various diseases, there has been an ongoing effort to aggregate and harmonise WES and WGS data from large-scale disease and population projects and to make them publicly available as a reference variome. This started in 2012 with a funder project called Exome Aggregation Consortium (ExAC) which harvested WES data from over 60,000 individuals; this was followed by The Genome Aggregation Database (gnomAD), of which three versions have been released so far, covering 71,702 genomes from unrelated individuals aligned against GRCh38 (v3). Also, in 2012, came the announcement of the 100,000 genomes project by the UK Government. The project sequenced the genomes of 100,000 NHS patients with particular focus on those with rare disease(s) and cancer. Another useful resource is the Trans-Omics for Precision Medicine (TopMed) program, which aims to sequence over 120,000 well-phenotyped individuals as well as collect other omics datasets. The assertion of ethnic diversity is an important consideration, and several initiatives such as KoVariome [82], Genes and Health [83] and BioBank Japan [84] are addressing this issue. Furthermore, new disease cohort genomic databases are being established. Examples include: NIHR BioResource Rare Disease Study (NBR) [77], Inflammatory Bowel Disease BioResource [85], Genetic Links to Anxiety and Depression (GLAD) [86] and Eating Disorders Genetics Initiative (EDGI) [87]. The advantages of these datasets are numerous; first, they provide allele frequencies for diverse populations, second, they help to address the overestimation of disease penetrance arising from the historical focus on multiplex pedigrees [88], and third, through acknowledging variable penetrance, they help to identify genetic and environmental disease modifiers [89].

#### 4.2.4. Reference Genome

The Human Genome Project was completed in 2003 [90,91] and since then, successive iterations of the human reference genome have been published, updated, and refined by the Genome Reference Consortium (GRC). Recent versions include GRCh37 (hg19) and GCRh38 (hg38) released in 2009 and 2013, respectively. These are both composite genomes, i.e., derived from the sequence of several anonymous donors; the make-up of these two assemblies is largely similar, with approximately 93% of the primary assembly composed of sequences from 11 genomic clone libraries.

To date, most large-scale PAH studies have aligned their data to the GRCh37 reference genome (Table 2), with only a couple of recent studies aligning their data to GRCh38 [25,55]. Even though GRCh38 was released seven years ago, the transition from GRCh37 to GRCh38 has been a long process and recent analyses have sought to compare the two reference panels. Guo et al. [92] demonstrated that GRCh38 provides a more accurate analysis of human sequencing data due to the improved annotation of the exome and the additional reads aligned to GRCh38, findings which indicate better structural and sequence representation. In addition, Pan et al. [93] noted that GRCh38 had better genome coverage, with a 5% increase in the number of SNVs identified. In comparison to GRCh37, GRCh38 altered 8000 nucleotides, corrected several misassembled regions, filled in gaps, and increased the number of genes and protein-coding transcripts [92]; additionally, GRCh38 is the first human reference genome to contain sequence-based representations for the centromeres [94]. Whilst GRCh37 is a single representation of multiple genomes, with only three regions containing alternative sequences (UDP-glucuronosyltransferases 2B subfamily (*UGT2B*) on chromosome 4, the major histocompatibility complex (MHC) region on chromosome 6, and the *MAPT* gene on chromosome 17) [95], GRCh38 includes 261 alternate loci across 178 genomic regions, providing a more robust representation of human population variation [94]. The increased level of alternative sequence representation requires new analysis methods to support their inclusion yet at present, most tools and pipelines do not make use of these [95]. Despite the advantages, GRCh38 still contains gaps and errors at repetitive and structurally diverse regions [96]. Additionally, as it is a mosaic haploid representation of the human genome [94], in which poor alignment can affect the detection of alleles in regions of high variation, such as the MHC locus and KIR, it is unlikely to truly represent human diversity [96]. It does, however, provide the starting point for a more inclusive population-based reference genome, or pan-genome [97] and as such, will play an evolving role in the generation of individual diploid genome assemblies and graph-based representations of genome-wide population variation [98,99,100], thereby providing unique opportunities for data analysis.

### 4.3. Studies

#### 4.3.1. Rare Genetic Variation

Over the last two decades, forward genetics approaches have associated PAH with numerous genes (Table 2); the level of evidence supporting the causal role of these genes, however, is variable and depends on multiple factors (Table 3). PAH is considered to be a monogenic condition transmitted in autosomal dominant fashion with incomplete penetrance. Heterozygous germline mutations in *BMPR2*, a member of the TGF-β superfamily, are the most common genetic cause of PAH [101,119], accounting for over 80% of familial PAH, and approximately 25% of idiopathic PAH [24]. Additional mutations within the TGF-β/BMP signalling pathway, such as activin A like type 1 (*ACVRL1*), endoglin (*ENG*), SMAD family members (*SMAD1*, *SMAD4*, *SMAD9*), caveolin 1 (*CAV1*), growth differentiation factor 2 (*GDF2*), loss of function variants in channelopathy genes, potassium two pore domain channel subfamily K member 3 (*KCNK3),* ATP binding cassette subfamily C member 8 (*ABCC8*), ATPase type 13A3 (*ATP13A3*), variants within developmental transcription factors, SRY-box transcription factor 17 (*SOX17*) and T-box transcription factor 4 (*TBX4*), and newly reported risk genes, γ-glutamyl carboxylase (*GGCX*), kallikrein 1 (*KLK1*) [25], kinase insert domain receptor (*KDR*) [30], fibulin 2 (*FBLN2*) and platelet-derived growth factor D (*PDGFD*) [118], have all been identified as individually rare causes of PAH. Infrequent cases of autosomal recessive transmission in *KCNK3* [120] and *GDF2* [121] have been associated with early disease onset and severe phenotype. A subtype of PAH, PVOD/PCH is linked to biallelic mutations in *EIF2AK4* [109].

##### Autosomal Dominant Mode of Inheritance

TGF-ß Pathway

In 2000, the International Primary Pulmonary Hypertension (PPH) Consortium demonstrated that familial PAH (FPAH) is caused by mutations in *BMPR2*, located on chromosome 2, encoding a TGF-β type II receptor [101]. They established a panel of eight kindreds, in which at least two members had the typical manifestations of PAH. Sequence variants were detected in seven probands; these variants, including two frameshift, two nonsense and three missense mutations, were distributed across the gene and each of the amino acid substitutions occurred at a highly conserved and functionally important site of the *BMPR2* protein. They observed segregation of the mutations with the disease phenotype in seven of the eight families studied. As control subjects, they screened 150 normal chromosomes from the same population and 64 normal chromosomes from ethnically diverse subjects and observed no *BMPR2* mutations [101]. The predicted functional impact of these mutations, their segregation with the phenotype, and the absence of these variants in healthy controls provided strong support for the role of *BMPR2* and the TGF-β signalling pathway in the pathobiology of PAH. The role of *BMPR2* mutations has been subsequently reported in IPAH. Thomson et al. [102] investigated *BMPR2* gene mutations in 50 unrelated IPAH patients with no family history of the disease. In 13 patients (26%), 11 novel heterozygous mutations in *BMPR2* were identified, these included three missense, three nonsense and five frameshift. They also sequenced both parents for five of the 13 probands; paternal transmission was observed for three families, whereas the remaining two mutations arose spontaneously. *BMPR2* mutations were not observed in 150 normal chromosomes [102]. Screening of other disease subtypes revealed *BMPR2* mutations among patients with PAH associated with congenital heart disease (PAH-CHD) [175] and PVOD [176].

Large cohort studies have proved useful in defining the relative contribution of *BMPR2* mutations in various PAH subtypes. Gräf et al. [24] reported rare heterozygous *BMPR2* mutations in 160 of 1048 PAH cases (15.3%); the frequency of *BMPR2* mutations in FPAH, IPAH and anorexigen-exposed PAH were 75.9%, 12.2% and 8.3%, respectively. Fourteen percent of *BMPR2* mutations resulted in the deletion of larger protein-coding regions, ranging from 5 kb to 3.8 Mb in size. Additionally, 52% of the observed *BMPR2* mutations were newly identified in their study [24], suggesting that nearly two decades after the first *BMPR2* mutation was identified, the use of WGS has allowed for closer study of *BMPR2*, including large deletions around the *BMPR2* locus, and the TGF-β pathway. Another large-scale study in a more heterogeneous group of patients (Group 1 PAH) [25] reported *BMPR2* mutations in 180 of 2572 cases (7%); the frequency of *BMPR2* mutations in FPAH and IPAH patients were 62.4% and 9.3%, respectively. Taken together, over 600 distinct mutations in *BMPR2* have been identified in PAH patients [24,25,177,178,179] of which around 70–80% are identified in FPAH and 10–20% in IPAH [180].

Importantly, impaired *BMPR2* signalling was shown to be a universal feature of PAH and pointed towards other key members of the canonical *BMPR2* signalling pathway as potential culprits for the disease [181,182,183].

Mutations in *ACVRL1* and *ENG* have been reported in PAH patients and in patients with PH in association with hereditary hemorrhagic telangiectasia (HHT). HHT is a rare autosomal dominant genetic disorder characterised by arteriovenous malformations and multiple telangiectasias [184]; it is frequently linked to defects in *ACVRL1* and *ENG* and as HHT and PAH may co-present in families, suggests a common molecular aetiology [103,104,105]. Of note, PH secondary to high cardiac output from arteriovenous fistulas is much more common in HHT, and such phenocopies, if unrecognised, may introduce significant bias to the studies [185]. Conversely, I/HPAH associated with *ACVRL1* and *ENG* can occur without clinical features of HHT [105,185,186], as the latter shows age-related penetrance. In a large case-control study, which employed deep phenotyping prior to association analysis, *ACVRL1* was associated with HPAH [30] but fell just below the cut-off for significance when studied in unselected patients with Group 1 PAH [25].

Two studies using targeted sequencing of *BMPR2* signalling intermediates provided further evidence supporting the role of this pathway in the pathogenesis of PAH. Shintani et al. [106] screened 23 patients with IPAH for mutations in *ENG*, *SMAD1*, *SMAD2*, *SMAD3*, *SMAD4*, *SMAD5*, *SMAD6* and *SMAD9* (*SMAD8*) and identified a nonsense mutation in *SMAD9* in a child who was diagnosed at eight years of age and his unaffected father. The results of immunoblotting and co-immunoprecipitation assays indicated that the *SMAD9* mutant disturbs the downstream signalling of TGF-β/BMP. In a later study, *SMAD1*, *SMAD4*, *SMAD5* and *SMAD9* were screened by direct sequencing in a cohort of 324 PAH cases (188 IPAH and 136 anorexigen-induced PAH) [54]. Four gene defects in three genes were observed. A novel missense variant in *SMAD1* was observed in an IPAH patient, a predicted splice-site mutation and a missense variant in *SMAD4* were observed in two IPAH patients and a novel missense variant in *SMAD9* was observed in a patient of Japanese origin. These four variants were absent in the 960 European and 284 French control samples and the *SMAD9* variant was excluded from the panel of 340 Japanese controls. A case-control study using WGS detected two cases harbouring protein-truncating variants in *SMAD1*, of which one co-existed with a protein-truncating variant in *BMPR2*, and eight *SMAD9* variants, two of which co-occurred with protein-truncating variants in *BMPR2* and *GDF2*; statistical analysis did not reveal significant association with studied phenotypes [30]. In another large cohort study (*n* = 2572 cases, 72% European), deleterious variants were observed in *SMAD1* (two cases), *SMAD4* (two cases) and *SMAD9* (13 cases) but were not statistically significant [25]. Taken together, these findings demonstrate that variations within the *SMAD* family have a small effect size, suggesting that a second genetic or environmental hit is needed, or that they perturb other non-investigated pathways.

Besides *BMPR2* mutations, *CAV1* mutations are a rare cause of PAH. Variants in *CAV1* were initially implicated in PAH pathogenesis by exome sequencing of a three-generation family with autosomal dominant HPAH who were negative for established variants in the TGF-β family [74]. They identified a frameshift mutation in *CAV1*; all PAH patients and several unaffected family members carried the *CAV1* mutation, suggesting incomplete penetrance. Subsequent evaluation of an additional 62 unrelated HPAH and 198 IPAH patients identified an independent de novo *CAV1* mutation in a child with IPAH. Two separate studies have since identified *CAV1* mutations in PAH patients; the first identified a novel heterozygous frameshift mutation in an adult PAH patient with a paediatric-onset daughter who died at nine years old [140], and the second identified deleterious variants in 10 patients with I/F/APAH, with three related cases carrying the same likely gene damaging mutation [25]. WGS in an I/HPAH cohort did not detect deleterious variants in *CAV1* [24]. Whilst these findings highlight the importance of caveolae in the homeostasis of the pulmonary vasculature, the link between *CAV1* mutations and PAH requires further study.

Bone morphogenetic protein (BMP)-9 (encoded by *GDF2*) and *BMP10* are ligands involved in TGF-β signalling pathway. Wang et al. [121] identified a novel homozygous nonsense mutation in the *GDF2* gene in a five-year-old Hispanic child with severe PAH. Genetic testing revealed that both parents were heterozygous for the same mutation, indicating that the child inherited the *GDF2* mutant allele from each parent. This study was the first to report a novel homozygous nonsense mutation in *GDF2* in an IPAH patient, suggestive of the causative role of *GDF2* mutations in PAH. Further evidence came from the NBR study which identified associations between rare heterozygous missense (*n* = 7) and frameshift variants (*n* = 1) in adult-onset IPAH (88% European) [24]; additionally, Hodgson et al. [115] identified two patients with large deletions encompassing the *GDF2* locus and several neighbouring genes.

The identification of *GDF2* mutations has since been independently replicated in a Chinese cohort [26]. Wang et al. [26] performed an exome-wide gene-based burden analysis on two independent case-control studies. The discovery analysis, containing 251 IPAH patients, identified rare heterozygous mutations in *BMPR2* (49 cases), *ACVRL1* (15 cases), *TBX4* (10 cases), *SMAD1* (two cases), *BMPR1B* (one case), *KCNK3* (one case) and *SMAD9* (one case). In a gene-based burden analysis (cases: *n* = 251; controls: *n* = 1884), only three genes (*BMPR2*, *GDF2* and *ACVRL1*) had an exome-wide significant enrichment of mutations in IPAH cases when compared to healthy controls. *GDF2* mutations were identified in 17 cases (6.8%) and ranked second to *BMPR2* (56 cases, 22.3%). To validate the risk effects of *GDF2*, they performed WES in an independent replication cohort of 80 IPAH cases and in a second gene-based burden analysis (cases: *n* = 80; controls: *n* = 8624), *BMPR2*, *GDF2* and *ACVRL1* were again identified as the top three disease-associated genes. Within this analysis, five additional *GDF2* heterozygous mutations were identified. Among the 331 IPAH patients, they identified 22 cases carrying 21 distinct rare heterozygous mutations in *GDF2*, only two of which had been reported previously [24], accounting for 6.7% of IPAH cases. An independent cohort confirmed the genome-wide association of *GDF2* among 1832 PAH and 812 IPAH cases of European ancestry [25]; twenty-four *GDF2* variants were observed in 28 cases, only two of which had been reported previously, and 75% of these occurred in IPAH cases.

Additionally, gene panel sequencing of 263 PAH patients (180 IPAH, 11 FPAH, 13 drug and toxin-induced PAH and 59 sporadic PVOD) revealed two (1.2%) *BMP9* mutations in adult PAH cases [114]; due to the close similarity of *BMP9* and *BMP10* (a close paralogue of *BMP9* that encodes an activating ligand for *ACVRL1*), the *BMP10* gene was also included in the capture design. Two mutations were identified in *BMP10*, a truncating mutation and a predicted loss of function variant were identified in two severely affected IPAH patients. Two rare missense variants in *BMP10* were identified in patients with IPAH in an independent cohort [115]. These results emphasise the role of *GDF2* in the pathobiology of PAH and suggest *BMP10* might act as a predisposing risk factor.

Channelopathies

Channelopathies are a group of diseases caused by dysfunction of ion channels localised in cellular membranes and organelles. These diseases include, but are not limited to, cardiac, respiratory, neurological and endocrine disorders. LoF variants in channelopathy genes have also been reported in PAH.

The first channelopathy described in PAH was caused by a genetic defect in *KCNK3* in patients with familial and sporadic PAH [107]. *KCNK3* belongs to a family of mammalian potassium channels and encodes for a two-pore potassium channel which is expressed in pulmonary artery smooth muscle cells (PASMCs); this channel plays a role in the regulation of resting membrane potential and pulmonary vascular tone and vascular remodelling [120]. In studying a family in which multiple members had PAH, Ma et al. [107] identified a novel heterozygous missense variant in *KCNK3* as a disease-causing candidate gene within the family. WES was used to study an additional 10 probands with FPAH and two novel heterozygous *KCNK3* variants were identified and segregated with the disease. In addition, three novel *KCNK3* variants were identified in 230 patients with IPAH. These five variants were predicted to be damaging. In summary, *KCNK3* mutations were identified in three of 93 unrelated patients (3.2%) with FPAH and in three of 230 patients (1.3%) with IPAH [107].

In another study using targeted sequencing, two *KCNK3* mutations were observed in three patients from two families. One of these mutations, a homozygous missense variant in *KCNK3*, was identified in a patient belonging to a consanguineous Romani family; his affected mother and asymptomatic father were carriers of the same *KCNK3* mutation. This is the first report of a young patient with severe PAH carrying a homozygous mutation in *KCNK3* [120]. Of note, in the Ma et al. [107] study, as the pedigree suggested an autosomal dominant mode of inheritance, homozygous variants were excluded from the analysis. The two biggest case-control studies, reported by Gräf et al. [24] and Zhu et al. [25], identified heterozygous KCNK3 mutations in only four (0.4%) and three (0.1%) cases, respectively, and did not show statistically significant associations.

Conversely, Gräf et al. [24] detected statistically significant enrichment of rare deleterious variants in two new channel genes: *ATP13A3* and *AQP1*. Utilising a rigorous case-control comparison using a tiered search for variants, they searched for high-impact protein-truncating variants (PTVs) overrepresented in cases and identified a higher frequency of PTVs in *ATP13A3* (six cases). *ATP13A3* is a poorly characterised member of the P-type ATPase family of proteins that transport a variety of cations across membranes [187]; *ATP13A3* is thought to play a role in polyamine transport [188]. Within PAH cases, Gräf et al. [24] identified three heterozygous frameshift variants, two stop gain, two splice region variants and four heterozygous likely pathogenic missense variants in *ATP13A3*. These variants were predicted to lead to loss of ATPase catalytic activity, and to destabilise the conformation of the catalytic domain; six variants were predicted to cause protein truncation, suggesting that loss of function of *ATP13A3* contributes to PAH pathogenesis.

Within the same study, SKAT-O analyses revealed a significant association with rare variants in *AQP1* [24]; *AQP1* ranked second in their combined rare PTV and missense variant case-control analysis. Along with statistical evidence, familial segregation of *AQP1* variants with the phenotype was shown in three families [24]. *AQP1* belongs to the aquaporins family, a family of water-specific membrane channel proteins that facilitate water transport in response to osmotic gradients [189]. Zhu et al. [25] identified seven cases with *ATP13A3* rare deleterious variants. However, *ATP13A3* and *AQP1* failed to reach genome-wide significance in their study and additionally, *AQP1* was not among the expanded list of genes with *p*≤ 0.001 for either the whole cohort or the IPAH subset.

Interestingly, a mutation in ATPase Na+/K+ transporting subunit α 2 (*ATP1A2*), previously associated with familial hemiplegic migraine (FHM), was reported in a 24-year old male with IPAH and history of FHM [190]. Genetic analysis of the proband and two siblings (one with FHM) revealed a nucleotide substitution in the coding sequence of the *ATP1A2* gene for both the proband and the affected sibling [190]. A co-occurrence of PAH and FHM supports the hypothesis of a potential common pathophysiological link; several studies have reported the presence and activity of the α2-subunit of the Na⁺/K⁺-ATPase in pulmonary vascular smooth muscle cells [191,192], and the decrease in expression and/or activity of different types of K⁺ channels in PASMCs of IPAH patients [193,194], and Montani et al. [190] suggested that mutations in *ATP1A2* may contribute to pulmonary arterial remodelling through the disturbance of intracellular Ca^2^⁺ and K⁺ concentrations.

Mutations in *ABCC8* have recently been identified as a potential second potassium channelopathy in PAH [113]. *ABCC8* encodes SUR1 (sulfonylurea receptor 1), a regulatory subunit of the ATP-sensitive potassium channel; *ABCC8* is highly expressed in the human brain and endocrine pancreas and moderately expressed in human lungs [195]. Mutations in *ABCC8* have previously been related to type II diabetes mellitus and congenital hyperinsulinism [196]. In exome-sequencing a cohort of 99 paediatric- and 134 adult-onset Group 1 PAH patients (182 IPAH and 52 HPAH), Bohnen et al. [113] identified a de novo heterozygous predicted deleterious missense variant in *ABCC8* in a child with IPAH. All individuals within this cohort and the second cohort of 680 adult-onset PAH patients (NBR study) were screened for rare or novel variants in *ABCC8*. Eleven heterozygous predicted damaging *ABCC8* variants were identified, seven of these were observed in the original cohort, including one familial case. In a study of 2572 PAH cases, Zhu et al. [25] identified rare deleterious variants in newly reported risk genes and nearly two-thirds of these variants were in *ABCC8* (26 variants in 29 IPAH/APAH patients). In a more recent study, utilising a custom NGS targeted sequencing panel of 21 genes, Lago-Docampo et al. [197] identified 11 rare variants in *ABCC8* within a cohort of 624 paediatric and adult patients from the Spanish PAH registry. To date, *ABCC8* variants have been identified in patients with IPAH, FPAH, PAH-CHD and APAH and account for ~0.5–1.7% of cases.

Transcription Factors

An emerging category of HPAH is the one that can be labelled as a disorder of transcriptional regulation. Interestingly, variants within developmental transcription factors are enriched in paediatric patients.

The first transcription factor implicated in the pathogenesis of PAH was *TBX4*. *TBX4*, expressed in the atrium of the heart, limbs, and the mesenchyme of the lung and trachea, encodes a transcription factor in the T-box gene family [198]. Deletions and LoF mutations in *TBX4* cause a variety of developmental lung disorders [199] and have been identified as a prominent risk factor in small patella syndrome (SPS) [200], childhood-onset PAH [108] and more recently, persistent pulmonary hypertension in neonates [201]. In a 2013 study incorporating array-comparative hybridisation and direct sequencing, three *TBX4* mutations and three novel *TBX4* microdeletions were detected in six out of 20 children with I/HPAH and interestingly, features of SPS were detected in all living *TBX4* mutation carriers [108].

Zhu et al. [140] performed exome sequencing on a cohort of 155 paediatric- and 257 adult-onset PAH patients. Within 13 probands (12 paediatric- and one adult-onset), they identified 13 likely pathogenic/predicted highly deleterious *TBX4* variants; eight of these variants were inherited from an unaffected parent, whereas one was de novo. This pattern is consistent with the incomplete penetrance observed for *BMPR2* mutation carriers [122]. Similar frequencies of rare, deleterious *BMPR2* mutations were observed in paediatric- and adult-onset I/FPAH patients; however, there was significant enrichment of rare, predicted deleterious *TBX4* mutations in paediatric- (10 of 130 patients) compared with adult-onset (0 of 178 patients) IPAH patients. In comparison to *BMPR2* mutation carriers, *TBX4* carriers had a 20-year younger age of onset, with a mean age of onset of 28.2 ± 15.4 years and 7.9 ± 9.0, respectively. After *BMPR2* mutations (10%), variants in *TBX4* (7.7%) conferred the highest degree of genetic risk of paediatric-onset IPAH [140]. Similar estimates of *BMPR2* (12.5% of I/FPAH patients) and *TBX4* mutation carriers (7.5% of I/FPAH patients) were observed in a study of 66 paediatric patients [141]. Additionally, in a 2019 study, examining 263 PAH and PVOD/PCH patients (paediatric and adult cases), *TBX4* mutations were the second most frequent mutations after *BMPR2* in both paediatric and adult cases [114].

Indicative of bimodal age distribution, pathogenic *TBX4* variants have also been reported in adult-onset PAH. Gräf et al. [24] identified deleterious heterozygous rare variants in 14 cases, Navas Tejedor et al. [120], in a Spanish cohort of 136 adult-onset PAH patients, identified three pathogenic mutations, and Kerstjens-Frederikse et al. [108], in a much smaller adult cohort (*n* = 49), detected a rare *TBX4* mutation. In a more recent study, 448 index patients were screened for PAH predisposing genes; 20 patients (nine childhood-onset) from 17 unrelated families carried heterozygous mutations in the *TBX4* gene, bringing the frequency of *TBX4* mutations in France to 6% and 3% in childhood- and adult-onset PAH, respectively [142]. Within this cohort, SPS was present in 80% of cases [142] and interestingly, all patients showed decreased DLCO and 87% had parenchymal abnormalities. These findings suggest that *TBX4* mutations may occur with or without skeletal abnormalities and whilst such mutations are mainly associated with childhood-onset PAH, the prevalence of PAH in adult *TBX4* mutation carriers could be up to 3%, depending on the population studied.

A recent study of 1038 PAH patients observed that rare heterozygous variants in *SOX17* were significantly overrepresented in the I/HPAH cohort [24]. *SOX17* is a member of the conserved SOX family of transcription factors. These transcription factors play a pivotal role in cardiovascular development and figure prominently in the aetiology of human vascular disease; they are involved in the regulation of embryonic development, the determination of cell fate and participate in vasculogenesis and remodelling [202]. In the NBR study [24], deleterious variants in *SOX17* were detected in less than 1% of the studied population and were characterised by younger age at diagnosis; familial segregation was shown in one patient.

To identify novel genetic causes of PAH-CHD, Zhu et al. [75] performed WES in 256 PAH-CHD patients; the cohort included 15 familial and 241 sporadic cases. Fifty-six percent of the cohort were of European ancestry and 26% were Hispanic; most cases (56%) had an age of onset < 18 years and so were categorised as paediatric-onset. The cohort was screened for 11 known risk genes for PAH and 253 candidate risk genes for CHD; PAH risk variants were identified in only 6.4% of sporadic PAH-CHD cases and four of the 15 familial cases. They performed a case-control gene-based association test of rare deleterious variants comparing European cases and controls (gnomAD: *n* = 7509 non-Finnish Europeans) and identified *SOX17* as a novel PAH-CHD candidate risk gene [75]. They estimated that rare deleterious variants in *SOX17* contributed to approximately 3% of European PAH-CHD patients. Following this discovery, they screened for *SOX17* variants in non-European cases and an additional cohort of PAH patients without CHD (*n* = 413) and identified five additional rare variants in the PAH-CHD cohort and three additional rare variants in the I/HPAH cohort [75]. In this second cohort, rare deleterious variants in *SOX17* were observed in 0.7% of cases. In total, 13 patients across the two cohorts were observed to have rare deleterious *SOX17* variants; nine of these had paediatric-onset PAH, suggesting these variants may be enriched in paediatric patients.

A Japanese study also demonstrated familial segregation of *SOX17* variants [112]. This study whole-exome sequenced 12 patients with PAH, 12 asymptomatic family members and 128 index cases and identified *SOX17* mutations in four PAH patients (three of these had congenital heart defects, i.e., atrial septal defect or patent ductus arteriosus) and one asymptomatic family member. Interestingly, the same heterozygous missense mutation in *SOX17* (c.397C) was observed in a Japanese patient [112] and in a patient with PAH from the NBR study [24], suggesting that this base position may be a pan-ethnic mutational hot spot. Taken together, these data strongly implicate *SOX17* as a new risk gene for PAH-CHD and suggest that this gene has a pleiotropic effect. Replication analyses in other PAH cohorts, with specific PAH subclasses, are needed to confirm the precise role of *SOX17*.

New Genes

In recent years, new risk genes for PAH have emerged from large WES and WGS studies. Zhu et al. [25] performed targeted gene sequencing alongside WES in a large cohort from the National Biological Sample and Data Repository for PAH (US PAH Biobank: *n* = 2572). Despite screening for 11 established PAH risk genes and seven recently reported risk genes, they failed to identify rare deleterious variants in known risk genes for 86% of the PAH Biobank cases (Group 1 PAH). They performed gene-based case-control association analysis and to prevent confounding by genetic ancestry, only participants of European ancestry (cases: *n* = 1832; controls: *n* = 7509 gnomAD WGS subjects and *n* = 5262 unaffected parents from the Pediatric Cardiac Genomics Consortium) were included. Using a variable threshold method, they identified two genes that exceeded the Bonferroni-corrected cut-off for significance: *BMPR2* and *KLK1*. *KLK1*, which encodes kallikrein 1, also known as tissue kallikrein, has not previously been associated with pulmonary hypertension.

The analysis was then repeated using 812 European IPAH cases and significant associations were observed for *BMPR2*, *KLK1* and *GGCX*. *GGCX* encodes γ-glutamyl carboxylase and has previously been implicated in coagulation factor deficiencies and vascular calcification [203], but again, never in PAH. In a final analysis, the entire PAH Biobank cohort was screened for rare deleterious variants in *KLK1* and *GGCX*; twelve cases carried *KLK1* variants (all European), whereas 28 cases carried *GGCX* variants (19 European, six African, three Hispanic), accounting for ~0.4% and ~0.9% of PAH Biobank cases, respectively. Carriers of *KLK1* and *GGCX* had a later mean age of onset and relatively moderate disease phenotype compared to *BMPR2* carriers. Both *KLK1* and *GGCX*, expressed in the lung and vascular tissues, play an important role in vascular haemodynamics and inflammation. Whilst Zhu et al. [25] identified *KLK1* and *GGCX* as new candidate risk genes for IPAH, suggesting new pathogenic mechanisms outside of the TGF-β/BMP signalling pathway, further research needs to be conducted to better understand these findings, especially in larger cohorts of similar phenotypic characteristics.

In a recent large-scale analysis utilising the 13,037 participants enrolled in the NBR study, of which 1148 patients were recruited to the PAH domain, we discovered *KDR* as a novel PAH candidate gene [24,30] utilising the Bayesian model comparison method, BeviMed [64], and deep phenotype data. Under an autosomal dominant mode of inheritance, high impact variants in *KDR* were associated with a significantly reduced KCO (transfer coefficient for carbon monoxide) and older age at diagnosis [30]. Six ultra-rare high impact variants in *KDR* were identified in the study cohort; four of these were in unrelated PAH cases, one in a relative and one nonsense variant was identified in a non-PAH control subject. The latter variant appeared late in the protein sequence and hence might not impair protein structure. To seek further evidence for *KDR* as a new candidate gene for PAH, we analysed subjects recruited to two cohorts with similar phenotypic characteristics (US PAH Biobank: *n* = 2572; Columbia University Medical Center: *n* = 440); four additional individuals harbouring rare high impact *KDR* variants were identified, with one variant identified in both cohorts. A combined analysis of both cohorts confirmed the association of *KDR* with PAH. Further evidence came from a French cohort of 311 PAH patients prospectively analysed by targeted panel sequencing (which included *KDR*) [117]. Two index cases with severe PAH, from two different families, were found to carry LoF mutations in *KDR*, providing further genetic evidence for considering *KDR* as a newly identified PAH-causing gene. Across both studies [30,117], there are now three reported familial cases with a distinct phenotype in which LoF variants in *KDR* segregate with PAH and significantly reduced KCO.

Power to detect novel genotype–phenotype associations can be increased by merging existing datasets, aligning sequences to the most recent genome assembly, using the most up to date reference variome, and improved variant classification tools, as well as updated clinical phenotypes. Such an approach was recently taken by a large international consortium of 4241 PAH cases from three cohorts (US PAH Biobank: *n* = 2572; Columbia University Medical Center: *n* = 469; NBR: *n* = 1134). Of the available 4175 sequenced exomes, most cases (92.6%) were adult-onset, with 54.6% IPAH, 34.8% APAH, 5.9% FPAH and 4.6% other; 74.5% of the cohort was European [118]. Gene-based case-control association analysis in unrelated participants of European ancestry was performed, using 2789 cases and 18,819 controls taken from the Simons Foundation Powering Autism Research for Knowledge (SPARK) cohort and gnomAD, before screening the whole cohort, including non-Europeans, for rare deleterious variants in associated genes. Statistical analyses revealed that rare predicted deleterious variants in seven genes were significantly associated with IPAH, including three established PAH risk genes (*BMPR2*, *GDF2*, and *TBX4*), two recently identified candidate genes (*SOX17* and *KDR*) and two new candidate genes (*FBLN2* and *PDGFD*).

Both new candidate genes have known functions in vasculogenesis and remodelling but have not previously been implicated in PAH. In total, they identified seven cases with *FBLN2* variants and ten cases with *PDGFD* variants, accounting for 0.26% and 0.35% of IPAH cases, respectively; most of these were of European ancestry and all were adult-onset, except for one paediatric *PDGFD* variant carrier. Analysis of single-cell RNAseq data showed that *FBLN2* and *PDGFD* have similar expression patterns to well-known PAH risk genes [118]. Whilst this provides additional support and mechanistic insight for the new genes, as with the discovery of *KLK1* and *GGCX*, variants within *FBLN2* and *PDGFD* require independent validation.

##### Autosomal Recessive Mode of Inheritance

PVOD/PCH has recently been reclassified as an ultra-rare form of Group 1 PAH [18]. PVOD and PCH often show significant phenotypic overlap. Indeed, 73% of patients diagnosed with PVOD are found to have capillary proliferation and 80% of patients with PCH demonstrate typical venous and arterial changes [204], and are, therefore, referred to as PVOD/PCH. Clinically, PVOD/PCH is characterised by early-onset, significantly reduced DLCO and patchy centrilobular ground-glass opacities, septal lines and lymph node enlargement seen on high-resolution computed tomography. The disease outcome is dismal, with rapid progression and frequent pulmonary oedema in response to PAH medication. Similarly to PAH, PVOD/PCH can present as either a sporadic or familial disease [205,206]. It was the latter that triggered a family-based study into the genetic basis of this condition. Familial linkage mapping, WES, and Sanger sequencing were employed and identified biallelic *EIF2AK4* mutations in affected siblings. Subsequently, biallelic *EIF2AK4* mutations were also identified in 25% of sporadic PVOD cases [109], 11.1% of HPAH cases (one of nine cases) [110] and 1.04% of I/HPAH cases (nine of 864 cases) [111]. Harbouring *EIF2AK4* mutations confer a poor prognosis irrespective of clinical diagnosis and importantly, radiological assessments were unable to distinguish reliably between PVOD/PCH patients and patients with IPAH [111].

#### 4.3.2. Common Genetic Variation

Complex pathobiology, low penetrance, heterogeneous phenotype and variable disease trajectory allow for common sequence variation that contributes to PAH risk and natural history. In HPAH, several common genetic variants were shown to impact the disease. Firstly, *BMPR2* mutations are a significant source of sex-related bias in disease penetrance (42% in females vs 14% in males) [122]; one explanation for this could be gene polymorphisms involved in estrogen metabolism [207]. Females harbouring deleterious variants in *BMPR2* show a significant reduction in *CYP1B1* gene expression and as a consequence, a lower 2-hydroxyestrone to 16α-hydroxyestrone ratio leading to activation of mitogenic pathways [207]. Moreover, direct estrogen receptor α binding to *BMPR2* promoter results in reduced *BMPR2* gene expression in females and may contribute to the increased prevalence of PAH [208]. Secondly, variation in *BMPR2* expression in HPAH, caused by nonsense-mediated decay positive (NMD^+^) *BMPR2* mutations, is conditional upon individual polymorphisms of the wild type (WT) allele [209]. Thirdly, a SNP in *TGFβ1* modulates age at disease onset in patients harbouring *BMPR2* mutations [210].

In IPAH, a SNP (rs11246020) in the Sirtuin3 (*SIRT3*), mitochondrial deacetylase, in either homozygote or heterozygote fashion, was associated with increased acetylation of mitochondrial proteins compared to the IPAH patients or disease comparator group with the WT genotype [166]. *SIRT3*, by deacetylating and thus activating multiple enzymes and electron transport chain complexes, plays a significant role in cell bioenergetics, and its polymorphism has been associated with susceptibility to metabolic syndrome [211] and IPAH, but not APAH [166]. Similarly, uncoupling protein 2 (*UCP2*), shown to conduct calcium from the endoplasmic reticulum (ER) to mitochondria and suppressing mitochondrial function, has been implicated in the pathogenesis of PAH [168]. These findings have important therapeutic implications as common variants in *SIRT3* and *UCP2* predicted response to dichloroacetate, pyruvate dehydrogenase kinase inhibitor, in a PAH phase 2 clinical trial [167]. Renin–angiotensin–aldosterone (RAA) system has been implicated in the pathogenesis of both systemic and pulmonary hypertension. For example, polymorphisms in the gene encoding angiotensin-converting enzyme (ACE) have been associated with IPAH [212] and with diaphragmatic hernia with persistent pulmonary hypertension [213]. Likewise, common variation in angiotensin II type 1 receptor (*AGTR1*) was associated with age at diagnosis in PAH. These findings indicate that RAA might be a therapeutic target [34].

Considering the important role of vasoactive, angiogenic and angiostatic substances in the pathogenesis and therapy of PAH, studies investigating the impact of common variation on gene expression of these substances are warranted. Recently, Villar et al. [170] found a recurrent SNP (rs397751713) in the promoter region of the endothelin-1 (*END1*) gene that had important regulatory consequences in both IPAH and APAH patients; rs397751713, which consists of an adenine deletion, allows transcription factors (Peroxisome proliferator-activated receptor γ (PPARG) and Kruppel Like Factor 4 (KLF4)) to bind to the promoter. Both of these transcription factors are linked to PAH pathogenesis and have been suggested as potential therapeutic targets [214,215,216,217]. In a similar fashion, a polymorphism in a potent angiostatic factor, endostatin, Collagen Type XVIII α 1 Chain (*Col18a1*), was studied. A SNP (rs12483377) in *Col18a1* was observed at an increased frequency in PAH patients (MAF 21.6) relative to published controls (MAF 7.5), or patients with scleroderma without PAH (MAF 12.5). Rs12483377 encodes uncharged amino acid asparagine (N) at residue 104 in place of negatively charged aspartic acid (D); carriers of single minor allele A (genotype AG), had lower serum endostatin levels than those who carried WT (genotype GG) and showed better survival even after adjusting for confounders [173].

Along with studies looking at polymorphisms in a particular gene or its promoter region, two GWAS have been carried out in PAH. The first consisted of a discovery and validation cohort totalling 625 patients (*sans BMPR2* mutations) and 1525 healthy individuals. A genome-wide significant association was found at Cerebellin 2 Precursor (*CBLN2*) locus; rs2217560-G allele was more frequent in cases than in controls in both the discovery and validation cohorts with a combined odds ratio (OR) for I/HPAH of 1.97 [95% CI 1.59;2.45] (*p*-value 7.47 × 10^−7^). Although the rs2217560 genotype did not correlate with *CBLN2* mRNA expression in monocytes, higher *CBLN2* mRNA levels were seen in lung tissue explanted from PAH patients than in control lung samples. Further analysis showed that increasing concentrations of *CBLN2* lead to inhibition of proliferation of PASMCs [165].

To date, the largest PAH GWAS of multiple international I/HPAH cohorts, totalling 2085 cases and 9659 controls, identified two novel loci associated with PAH: an enhancer near *SOX17* (rs10103692, OR 1.80 (95% CI 1.55;2.08), *p* = 5.13 × 10^–15^), and a locus within *HLA-DPA1/DPB1* (rs2856830, OR 1.56 (95% CI 1.42;1.71), *p* = 7.65 × 10^–20^) [116]. CRISPR-mediated inhibition of the enhancer reduced *SOX17* expression; this finding corroborates and extends the previous discovery of the association of rare variants in *SOX17* with PAH [24]. The *HLA-DPA1/DPB1* rs2856830 genotype was not only associated with I/HPAH but also with a beneficial effect on survival (CC genotype 13.5 vs TT genotype 6.97 years) irrespective of baseline disease severity, age and sex [116]. The latter study did not replicate previous results at genome-wide significance.

### 4.4. Limitations, Challenges and Future Directions

The major limitations in PAH genetic studies originate from either their study design or the genetic methods used. The former is mostly limited by sample size and population structure and features in GWAS and RVAS alike, the latter is impacted by sequencing technology, genome assembly and analysis pipelines. Additionally, the variable quality of reporting may obscure the validity of the results. Several approaches may accelerate causal variant discovery. Firstly, international efforts to set up platforms and governance, thereby reducing the barriers to genotype and phenotype data harmonisation, across studies and countries, along with regulatory standards, compatible with the international legislative landscape, aid variant discovery. Secondly, the application of computational and/or intermediate phenotypes may further increase power to detect new risk genes. Likewise, researching isolated populations resulting from recent bottlenecks (i.e., Icelanders, Ashkenazi Jews) have shown promising results in other rare diseases. As for genetic methods, long-read technologies will improve significantly and with improvements to the error rate, long-read alignment to a pan-genome graph may identify additional genetic variants. Expansion of resources for allele frequency estimation (i.e., gnomAD), encompassing diverse ethnicities and variant classes, including structural variants, will improve variant interpretation. Studying gene sets that are likely to be enriched for disease-associated loci (i.e., implicated by GWAS, or expressed in disease relevant tissues) and extending RVAS to non-coding regions (a natural step given GWAS findings) is likely to further increase discovery yield.

## 5. Reverse Genetics

### 5.1. Concepts

Reverse genetic techniques have been used extensively in the context of PAH. These experimental approaches involve targeted modifications to specific genes in order to analyse phenotypic impact (Figure 1, process 3) and are critical in validating in silico predictions. Most deleterious mutations found in PAH patients are LoF, suggesting a knockout or knockdown approach to analysing the phenotypic effect of a mutation. The main benefit of direct gene mutagenesis is the permanence and potentially ubiquitous effect of that alteration. In comparison, gene knockdown is a transient change that can be created after the developmental stage in the animal life cycle. This is particularly useful for understanding LoF in genes that are embryonically lethal [123]. Conditional and inducible systems are also excellent means of circumventing this issue, permitting temporally and spatially controlled gene disruption [55,125,128,218]. Inducible systems display higher efficiency and limited side effects compared to stably-expressed mutations and have the added benefit of reversibility [219].

### 5.2. Methodology

Techniques for genetic modification have permitted targeted genetic analysis using in vitro and in vivo models of PAH. Modification of a specific gene, through mutagenesis or knockout, can be carried out using a wide range of tools, including Zinc-finger nucleases (ZFNs) [220], transcription activator-like effector nucleases (TALENs) [221], and CRISPR/Cas9 [222]. Similarly, transient genetic changes can be achieved through gene expression knockdown, such as RNAi [223], or using conditional/inducible expression systems, such as Cre/LoxP and Flp/FRT sites or the Tet inducible system [224]. Generating models with specific mutations is also critical for the identification of therapeutic targets and drug testing. Specific techniques and their utility in animal models have been extensively reviewed [224,225,226].

#### 5.2.1. In Vitro

As PAH is a disease of the pulmonary circulation, most studies seek to model genetic changes in physiologically relevant cell lines. This includes pulmonary artery endothelial cells (PAECs), PASMCs, blood outgrowth endothelial cells (BOECs) [227] and human umbilical vein endothelial cells (HUVECs). PAH is characterised by molecular and cellular deviations in pulmonary vascular function [228,229], including hyperproliferation [124,129,130,149], impaired migration [127,130,150] and aberrant apoptosis [124,230]. Traditionally, research has focused on using 2D cellular monolayers for analysis of these features; however, with rapidly evolving technologies, future analysis may utilise multi-layer, 3D models such as organoids [231] and hydrogels [232] to better replicate the pathobiology of PAH.

#### 5.2.2. In Vivo

Understanding the pathobiology of PAH is dependent upon robust animal models for functional analysis and exploration of potential therapeutics. To achieve this aim, a wide range of animal models have been developed across species, including pig, sheep, dog and most commonly rodents [233].

The most widely accepted rat models for PAH are pharmacologically induced, namely monocrotaline (MCT) and Sugen 5416-hypoxia (SuHx) models [234]. MCT is a toxin that induces damage to PAECs, causes RV and pulmonary arterial medial hypertrophy, and alters pulmonary artery pressures [235,236]. In the SuHx rat model, SU5416, a Vascular Endothelial Growth Factor Receptor (VEGFR) antagonist that promotes endothelial cell apoptosis and smooth muscle cell (SMC) proliferation [237], is administered followed by a period of hypoxia [234]. SU5416 requires hypoxia to induce severe PAH, causing vascular changes that model human IPAH [237]. Both models induce non-PAH related symptoms, including alveolar oedema, pulmonary vein occlusion, acute lung injury, liver toxicity and emphysema [238,239].

The chronic hypoxia mouse model is also widely used in PAH research, with mice housed at 10% oxygen for a variable length of time [234,240,241]. While mice display changes in pulmonary artery pressure, they display limited vascular remodelling, which consists of muscularization of vessels and medial hypertrophy of muscular resistance vessels [242]. Alternatively, pulmonary artery banding (PAB) has been used in various species to increase pulmonary artery pressures and induce RV hypertrophy. The main benefit of this technique is the lack of pharmacologic or environmental modification of the animal [243]. In practicality, the popularity of each model varies across PAH research groups; a recent poll showed that two-thirds of groups used two or more models [244], suggesting that no one model truly replicates the disease.

In comparison, genetic models tend to be more favourable as they are free from the systemic effects of pharmacological models. Several heterozygous *BMPR2* mutant models have been developed. While some models have shown elevated mPAP and PVR compared to WT littermates [123], others did not replicate these findings [245,246]. Alternate studies have shown the development of pulmonary vascular lesions [126] and often, an environment hit may be required for development of PAH [131,245,246]. More pronounced phenotypes are seen in homozygous deletions; as *BMPR2* is critical for embryogenesis, conditional deletion in pulmonary endothelial cells has been used and showed incomplete penetrance of the mutation, with a subset of mice displaying elevated RVSP, RV hypertrophy and inflammation [125].

The following genetic models have also been developed: *KDR* (cell-specific conditional deletion of *Kdr* in mice) [155], *AQP1* (*Aqp1* null mice [151] and *Aqp1* with a COOH-terminal tail deletion [149]), *TET2* (conditional, haematopoietic heterozygous and homozygous deletion in mice, generated using the Vav1-Cre system) [55], *BMPR1a* (several conditional knockout mice models) [156,157,159], *SOX17* (conditional deletion in mice, using *Demo1-Cre* in descendants of the splanchnic mesenchyme) [153], *UCP2* (knockout mouse model created using a plasmid vector carrying modified *Ucp2*) [169], *CAV1* (modified mouse cav-1 targeting construct) [138], *EIF2AK4* (modified mouse *Gcn2* targeting construct) [164], *SIRT3* (heterozygous and homozygous mice, generated using targeted embryonic stem cell clones harbouring a mutated *SIRT3*) [166], *ATP13A3* (generation of a protein truncating mutation in mouse *Atp13a3* using CRISPR/Cas9) [147], *SMAD9* (Disrupted *Smad8* allele generated in mice using LoxP/cre and Frt sites) [134], *KCNK3* (heterozygous and homozygous rats harbouring a CRISPR/Cas9-generated exonic, inframe deletion) [146] and *SMAD1* (mice with *Smad1* deletion in endothelial cells or SMCs, generated using L1Cre or Tagln-Cre, respectively) [136].

In humans, the severity of disease corresponds to the extent of pulmonary vascular changes and impairment of RV function. However, rodent models may display milder and isolated symptoms of PAH and, thus, require thorough characterisation to validate the model. Echocardiography is useful for the assessment of RV function but estimates of pulmonary pressures are inferior to direct measurements obtained through RHC [234]. Alternatively, cardiac magnetic resonance imaging is another noninvasive tool for assessment of RV function; it supersedes echocardiography in its ability to produce high quality, three-dimensional images from which the RV can be directly measured. However, it also has limited availability and bears higher running costs [247]. Close-chested RHC is preferred over open-chested RHC, as it is less invasive and preserves the negative intrathoracic pressure associated with breathing. Open-chested RHC is usually a terminal procedure, thus preventing longitudinal assessment of rodents; however, it also permits a more comprehensive assessment of haemodynamic characteristics and analysis using pressure–volume loops. Tissue harvesting permits further characterisation of the heart and lungs, including evaluation of RV hypertrophy [234].

It is important that measures are taken to recognise and limit bias in experimentation and phenotyping. This includes ensuring randomisation of groups when carrying out tests that require a control and treatment group, such as selecting mice to be pharmacological models or therapeutic treatment. Additionally, mice should be matched for all possible qualities, including strain, age and sex. During phenotype assessment, researchers should aim to avoid implicit bias by using blinded observers during all procedures. Furthermore, the data collected must be comprehensive and relevant for PAH, including RHC data and histological analysis [248].

### 5.3. Studies

#### 5.3.1. Rare Genetic Variation

##### Autosomal Dominant Mode of Inheritance

TGF-β Family

*BMPR2*, a receptor in the TGF-β pathway, is the most commonly mutated gene in PAH. BMP ligands induce signalling through binding with *BMPR2* and *ALK* 1, 2, 3 or 6 to induce downstream SMAD-mediated transcription [249]. This pathway plays important roles in cell proliferation, differentiation, migration, apoptosis and inflammation, which are often dysregulated in PAH. Alongside *BMPR2*, several pathway components have been implicated in disease; this includes the ALK and SMAD proteins, as well as coreceptors, such as endoglin [250]. Reverse genetics techniques have been used extensively on these genes to understand their role in PAH and their interaction with other members of the pathway. *BMPR2* has been extensively analysed using the methods described; under normal conditions it is highly expressed in vascular ECs, with lower levels of expression in SMC; however, reduced levels have been reported in the lungs of PAH patients [181]. In the endothelium, *BMP9* and *BMP10* induce binding of *BMPR2* and *ALK1*, along with co-receptor endoglin, to induce downstream signalling. Such signalling is often disrupted in HPAH patients that harbour mutations in *BMPR2* [251]. However, disruption of BMP-mediated signalling is also apparent in IPAH patients, despite the absence of *BMPR2* mutation [252].

Gene disruption and knockout techniques have been key to elucidating the role of *BMPR2* in pathogenesis. Such techniques have revealed dysregulation of endothelial nitric oxide synthase (eNOS) in cell culture models of PAH; siRNA-mediated knockdown of *BMPR2* induced endothelial dysfunction in human PAECs, caused by a reduction in BMP2/4-mediated production of the vasodilator nitric oxide by eNOS [127]. In addition, reduced *BMPR2* has been associated with overactive glycolysis [130] and hyperproliferation [124].

Furthermore, a key feature of PAH is incomplete penetrance in *BMPR2* mutation carriers. This has been replicated in mice carrying conditional knockout mutations of *Bmpr2* in pulmonary endothelial cells, with only a proportion of mice developing haemodynamic symptoms of PAH [125]. In addition, it has been shown that *BMPR2* mutations disrupt RV function, with heterozygous mice displaying impaired hypertrophy and lipid accumulation in the RV [128]. Moreover, EndoMT has been suggested to be a characteristic of PAH in both animal models and patients [253,254]. This includes a heterozygous *BMPR2* mutant rat model, which displayed overexpression of two molecular markers of EndoMT. Such changes were also seen in PAH patients when analysing mRNA and protein level changes in lung and artery tissue [129]. These in vivo models have therapeutic utility, having been used to test treatments that include rapamycin [129] and metformin [128] to alleviate symptoms.

The experimental techniques discussed have been highly important in identifying other members of the TGF-β pathway that are implicated in PAH pathogenesis. This includes the Smad proteins, for which functional evidence is limited. Functional analysis of variants detected in *SMAD1*, *SMAD4* and *SMAD9*, have shown impaired signalling in *Smad4* and *Smad9.* However, variants only have a moderate impact on phenotype, suggesting further genetic or environmental hits may be required to induce PAH [54]. Furthermore, analysis of *CAV1* mutations have revealed impaired protein trafficking in mouse embryonic fibroblasts carrying a disrupted copy of *CAV1*. Furthermore, patient fibroblast cells displayed reductions in both *CAV1* and associated accessory proteins [137]. Thrombospondin 1 (*THBS1*) mutants, which regulate TGFβ function, have been found to have impaired ability to activate TGFβ, and permit proliferation of PASMCs. Additionally, an electrophoretic mobility shift assay (EMSA) was used to analyse the binding of biotin labelled oligonucleotides with two transcription factors, MAZ and SP1, showing a reduction in efficiency [174].

As previously discussed, *GDF2*, encoding the ligand BMP9, is critical in *BMPR2*-mediated signalling in endothelial cells. Functional studies are particularly important in validating in silico predictions regarding the pathogenicity of certain variants. Hodgson et al. [115] analysed seven missense mutations in *GDF2*, which had been previously associated with PAH and predicted pathogenic in silico. When analysed in vitro, all variants displayed dysfunctional processing, secretion or protein stability. In addition, functional studies found a predicted benign mutation to be deleterious. Similarly, association analysis in IPAH patients of Chinese Han ethnicity identified *GDF2* mutations [26]. Six mutations were selected for functional studies, using mutant GDF2 plasmids transfected into HEK293E cells/PAECs. Mutants displayed reduced *BMP9* levels, reduced BMP activity and increased apoptosis [26].

There is some evidence implicating *BMPR1A*, which encodes the type I receptor ALK3, in PAH. A hypoxia mouse model harbouring a mosaic deletion of *Bmpr1a,* generated through conditional deletion of the gene in SMCs using the TRE-Cre system, displayed proximal artery stiffening caused by excess collagen. This may lead to constricted vessels and impairment of right ventricular function [255]. Furthermore, two independent studies have implicated reduced *Bmpr1a* expression with EndoMT and shown that the phenotype may be rescued by attenuating expression of TGFBR2, which is upregulated in the mice models [156,157]. Additionally, functional analysis of *BMPR1B* mutations found in PAH patients found changes in transcriptional activity, which resulted in increased SMAD8 phosphorylation [160].

Channelopathies

Alongside TGFβ-family members, various channel proteins have been linked to pathogenesis. Functional analysis was conducted on six mutations reported in *KCNK3*, all of which resulted in LoF. Patch clamp variants showed a reduction in current, with rescue of channel function, in a subset of variants by administration of a phospholipase inhibitor, ONO-RS-082 [107]. In addition, Bohnen et al. [145] introduced a heterozygous, single amino acid variant into *KCNK3* using site-directed mutagenesis and transfected hPASMCs with this mutant via a pcDNA3.1 plasmid. The physiological impact of this mutation was assessed using whole-patch clamp tests to observe changes to membrane potential and current, with experiments demonstrating loss of channel function in mutant cells compared to WT *KCNK3*. More limited evidence is available for the role of *ATP13A3*, a P-type ATPase, in PAH; siRNA-mediated knockdown of expression resulted in a reduction in cell proliferation and increased apoptosis, as well as the loss of endothelial integrity in hPAECs [230]. Furthermore, mutant *ATP13A3* mice, carrying a protein-truncating variant introduced using CRISPR/Cas9, demonstrated haemodynamic measurements indicative of PAH. This included reduced pulmonary artery acceleration time (PAAT) and increased RVSP, with rats also exhibiting reduced polyamine levels in their lungs [147].

*AQP1* has been associated with cell migration and vascularity. *AQP1* is highly expressed in microvascular endothelial cells, vascular smooth muscle cells and non-vascular endothelium [256], and has recently been proposed as a novel promoter of tumour angiogenesis [257]. A recent study demonstrated that depletion of *AQP1* reduced proliferation, migratory potential, and increased apoptosis of PASMCs [258]. Saadoun et al. [150] produced an *AQP1* knockout mouse model using targeted gene disruption; when implanted with melanoma cells, the developing tumours displayed impaired angiogenesis. In-depth analysis of *APQ1* function in endothelial cells, produced from mouse aortic cells, using cell migration, wound healing, proliferation and cord development assays, revealed critical roles of *AQP1* in cell migration. *AQP1* null cells showed reduced ability in all four assays. This was corroborated by the plasmid expression of *AQP1* and another water channel protein, *AQP4*, in non-endothelial cells, resulting in increased migration. Functional analysis has confirmed the role of *AQP1* in cell migration under hypoxic conditions. Yun et al. [149] used rat pulmonary-artery-derived PASMCs, infected with adenovirus vectors carrying either WT *AQP1* or *AQP1* with a 37 amino acid terminal deletion removing the COOH tail. Overexpression of *AQP1* increased levels of β-catenin and its downstream targets, including c-Myc and cyclin D1. However, loss of the COOH tail produced no such results, suggesting COOH-mediated upregulation of β-catenin. Both *AQP1* and β-catenin displayed elevated levels under hypoxic conditions. This was further corroborated by siRNA-mediated silencing of β-catenin, in cells subject to hypoxia, and consequent infection with adenoviral constructs. Cells were rescued from the migration and proliferation associated with hypoxic conditions.

Additionally, functional analysis has been conducted on *ABCC8*. Eight mutations were analysed in COS cells using patch-clamp tests and rubidium flux assays, as a measure of ion efflux. Loss of channel function was seen in all variants, with the rescue of function by the administration of the SUR1 activator drug, diazoxide [113]. Additionally, functional analysis of *KCNA5*, by overexpressing the gene in PASMCs, showed an increase in channel current that was rescued by administering either nicotine, bepridil, correolide or endothelin-1 [163]. Additional genes have been reported, but have limited functional evidence confirming their role in PAH. One such gene is *UCP2*, encoding uncoupling protein 2, which functions as a calcium channel in vascular mitochondria that mediates ion influx from the ER. Fluorescence resonance energy transfer imaging of mitochondria from resistance pulmonary arteries of mice harbouring a *Ucp2* knockout showed a reduced calcium concentration, impairing their function, and mimicking the effects of hypoxia. Impaired mitochondrial function has been associated with the hyperproliferative and anti-apoptotic symptoms seen in PH [168]. Another channel protein is TRPC6. Using Western blot analysis, TRPC6 was found to be upregulated in PASMCs obtained from IPAH patients, compared to control cells. The endothelin receptor antagonist, Bosentan, was found to act by downregulating *TRPC6* and had a more pronounced anti-proliferative effect on IPAH cells than on WT cells [259]. Alternatively, potassium channel function has been rescued using adenoviral infection of human Kv1.5, into mice exposed to chronic hypoxia for three to four weeks. Mice were alleviated of hypoxia-induced vasoconstriction, displaying reduced right ventricle and pulmonary artery medial hypertrophy, when compared to control mice subject to hypoxia [260].

Transcription Factors

TBX4 is an evolutionarily conserved transcription factor, which plays a critical role in limb development during embryogenesis [261]. Detection of a lung-specific *Tbx4* enhancer, specific to the lung mesenchyme and trachea during embryogenesis, suggest an important role in lung development [262]. It has been shown to induce myofibroblast proliferation and drive invasion of the matrix by fibroblasts, promoting the development of pulmonary fibrosis, and deletion of *Tbx4* has led to a reduction in fibrosis [262]. *TBX4* plays a role in lung development and has been linked to pulmonary fibrosis, suggesting a possible role in PAH; nonetheless, experimental evidence is required to confirm this. Similarly, *SOX17* has been shown to be critical for the development of pulmonary vasculature. A mouse model, harbouring a conditional deletion of *Sox17* in splanchnic mesenchyme descendants, displayed impaired pulmonary vascular development leading to death by three weeks of age [153]. Shih et al. [154] assessed the role of *SOX17* using siRNA-mediated knockdown of expression in PAECs, followed by angiogenesis assays that showed impaired vessel formation. Concurrently, a mutant *SOX17* cell line, generated through CRISPR/Cas9 mutagenesis of human iPSC, showed a reduction in expression of arterial genes compared to WT cells.

New Genes

More recently, a plethora of new genes have been found to harbour mutations in PAH patients; however, functional evidence for their involvement remains limited. *TET2*, involved in DNA demethylation, is a key epigenetic regulator and has been found dysregulated in PAH. Potus et al. [55] developed heterozygous and homozygous *Tet2* knockout mice models, by crossing floxed *Tet2* parents and *Vav1-cre* to excise the gene. Their phenotype was ascertained through echocardiography and RHC. Patients carrying *TET2* mutations were found to have a later age of onset compared to other PAH patients; this finding was replicated in mice. Genetic evidence has also implicated *KDR* in PAH; Winter et al. [155] developed a mouse model with a conditional deletion of the *KDR* gene, encoding VEGFR2, in endothelial cells. The mice developed mild PAH symptoms under normoxic conditions; however, these were exacerbated under hypoxia with mice displaying pulmonary vascular remodelling.

Other genes which have been associated with PAH through sequencing analysis have no functional evidence with respect to PAH; however, physiological functions reveal possible mechanistic roles in the disease. *GGCX* has also been associated with PAH using forward genetics studies but lacks any functional evidence to corroborate this. This gene encodes a protein that carboxylates glutamate residues on Vitamin-K-dependent proteins; these are vital for the activation of coagulation proteins [263], inhibiting vascular calcification and inflammation [25]. Mutations in *GGCX* have been associated with Vitamin-K-dependent clotting factors deficiency, a congenital bleeding disorder [264] and Pseudoxanthoma elasticum [265].

Additionally, *KLK1* plays a key role in cardiac and renal function, notably regulating blood pressure. Kinins are known to affect endothelial cells, especially playing a role in vasodilation, increasing vascular permeability, nitric oxide production and inflammation. Tissue kallikrein is highly expressed in the kidney, pancreas, central nervous system and blood vessels [266]. Kinins, which include bradykinin, are regulated by inactivating enzymes known as kininases, of which the ACE is the most well known [267]; ACE-inhibitors are effectively used to reduce blood pressure [268]. Indeed, hypertension has been associated with polymorphisms and deficiencies in *KLK1* expression in rat models [269,270]. While genetic variation in *KLK1* has been associated with essential hypertension in a Chinese Han cohort [271], other studies have reported conflicting evidence on its role in the condition [268]. Nonetheless, this gene and pathway offer potential therapeutic targets, having major functions in areas of dysfunction seen in PAH, demonstrated by the evidence showing that *KLK1* can be used for its vasodilative properties in the treatment of acute ischemic stroke [272].

##### Autosomal Recessive Mode of Inheritance

Anthony et al. [273] demonstrated that homozygous *Eif2ak4* knockout mice displayed increased levels of protein carbonylation, a symptom of oxidative damage that was induced by dietary leucine deprivation. Additionally, these mice exhibited reduced viability compared to controls, with some dying shortly after birth. Their findings suggest that the gene may elicit protective effects against oxidative damage.

#### 5.3.2. Common Genetic Variation

Another interesting PH animal model is the *Sirt3* knockout mice, which displays higher mPAP, PVR, RV hypertrophy and reduced exercise tolerance, compared with controls. This gene has been implicated in IPAH, with reduced expression in hPASMCs obtained from patients. The severity of symptoms in the model occur in a dose dependent manner, with *Sirt3* heterozygous mice displaying milder symptoms than homozygous knockout mice. This includes the extent of muscularisation and medial wall thickness of resistance PAs, which is more severe in homozygotes [166]. However, another *Sirt3*KO mouse model, against a C57BL/6 background [274] failed to develop PH, which may be attributed to differences between the mouse strains used. The 129/Sv strain used by Paulin et al. [166] possess a slower metabolism compared to C57BL/6 mice, which may mean loss of *Sirt3* is more damaging than in the latter strain.

Mutations in vasodilatory and vasoconstrictive proteins have been implicated in PAH, including *END-1*. In situ hybridisation has shown *END-1* upregulation in the lungs of patients with PAH, with greatest abundance in vessels subject to disease-associated remodelling [171]. Furthermore, analysis of a SNP found in the reporter region of *END-1* using a luciferase assay has shown upregulated activity. Homozygous mutants display a 30% increase in promoter activity and loss of binding of KLF4 and PPAR γ, which have both been linked to PAH [170].

Common variation in the DNA Topoisomerase II Binding Protein 1 (*TopBP1*) has been found in IPAH patients. Immunohistochemical analysis of vascular lesions demonstrated a reduction in TopBP1 protein. Analysis of cultured pulmonary microvascular endothelial cells from IPAH lungs, using q-PCR and Western blotting, showed a reduction in TopBP1 mRNA and protein. Further analysis of cells for increased susceptibility to DNA damage, by administering hydroxyurea and staining nuclei for phosphorylated histone foci, revealed an excess of DNA strand breaks and apoptosis. However, patient cells could be rescued by transfection with a plasmid containing wild-type human *TopBP1*, displaying a reduction in DNA damage-induced apoptosis [172].

In addition, functional analysis has been completed in *SOX17*. A luciferase reporter construct was generated for each haplotype of four heterozygous *SOX17* mutations and transfected into hPAECs to assess transcriptional activity [116,275]. Comparison of each haplotype revealed a haplotype-specific difference in promoter activity in a subset of variants. Two variants showed a statistically significant decrease in promoter activity in the risk allele, compared to the non-risk allele. Evidence exists for the intricate function of the various genes implicated in PAH, including regulatory roles of *SOX17*. VEGF has been shown to upregulate *SOX17*, while the Notch pathway plays critical roles in downregulating Sox17 in the endothelium. Additionally, there is a positive feedback loop between VEGR and *SOX17* [275]. These findings support the evidence that a wide range of mutations across genes are capable of eliciting similar phenotypic impacts through large-scale changes to pathway regulation.

### 5.4. Limitations, Challenges and Future Directions

To conclude, reverse genetics techniques are vital components of PAH research, without which forward genetics’ discoveries cannot be validated. Nonetheless, it is important that such studies are interrogated for any limitations that may affect their results. As previously mentioned, it is important to have a robust experimental design and protocol, which includes power calculations for animal studies, minimising bias and reporting all measurements in publications. Cellular models are powerful in vitro tools for understanding specific processes in PAH; however, focusing on certain cell types may produce more pronounced effects than those seen in vivo. Understanding the widespread impact of PAH mutations are best captured using animal models. However, it is important to remember that all fail to fully recapitulate the disease and that any toxic side effects of pharmacological models may confound results. Studies must choose models which are most appropriate for the hypothesis, or use multiple where necessary. Furthermore, focusing specifically on rodent models, these often capture longitudinal data through the use of terminal procedures for haemodynamic measurements. The experimental techniques discussed here are well established in the field; however, several new tools have been developed which may increase the speed and depth of information derived using reverse genetics; these include high-throughput technologies for mutagenesis and spatial gene expression analysis [276,277]. Such developments offer exciting prospects for future research in PAH.

## 6. Reverse Phenotyping

### 6.1. Concepts

The concept of reverse phenotyping (Figure 1, process 4) refers to the use of genetic marker data to refine the disease subgroup definitions [36]. The method was first used in phenotyping patients with sarcoidosis [278] and has proven successful in many other diseases characterised by high heterogeneity [279].

### 6.2. Theoretical Basis

Heterogeneity provides a unique challenge in the diagnosis and treatment of rare diseases, and is further complicated by the accuracy of reporting. Knowledge about the genetic makeup of the individual allows targeted treatments, thereby enhancing efficacy and decreasing the risk of potential side effects [280]. Conversely, knowledge about the phenotypic characteristics of mutation carriers is indispensable to match study design and methodological approaches to disease attributes [281].

### 6.3. Methodology

Studying the complexity of genotype–phenotype associations is of particular interest to medical genetics. Evidence for such studies can be retrieved from published peer-review literature and publicly available databases, and facilitated by computational tools such as the R package *VarformPDB* (Disease-Gene-Variant Relations Mining from the Public Databases and Literature), which captures and compiles the genes and variants related to a disease, a phenotype or a clinical feature from public databases including HPO (Human Phenotype Ontology), Orphanet, OMIM (Online Mendelian Inheritance in Man), ClinVar, and UniProt (Universal Protein Resource) and PubMed abstracts. Alternatively, the development of large biobanks that link rich electronic health record (EHR) data and dense genetic information have made it possible to enhance clinical characterisation of mutation carriers in a cost-effective fashion. Besides deep clinical, often longitudinal phenotyping, multi-omic approaches can be employed thanks to the concurrent collection of patients’ biological samples. A number of efforts have been made to facilitate EHR-based genetic and genomic research, such as the establishment of the UK Biobank [282], the Electronic Medical Record and Genomics Network [283] or the Precision Medicine Initiative Cohort Program [284], which allow both forward genetics and reverse phenotyping applications. Mining EHRs for genetic research purposes possesses a number of advantages over classical population or family-based genetic association studies, large samples can be collected over a short period of time in a cost-effective fashion. Moreover, the robustness of EHRs allows for the investigation of a spectrum of phenotypes in a hypothesis-free manner. Familial relationships reported in EHRs have also been used for the estimation of disease heritability shown to be consistent with the literature [285].

Multiple tools are being used by the Clinical Genetics community to address the issue of reverse phenotyping when interpreting NGS data. This includes DECIPHER [286], a publicly-available database that houses a collection of case-level evidence for 36,801 individuals with genetic diseases, for comparison of phenotypic and genotypic data. Projects affiliated to DECIPHER can deposit and share patients, variants and phenotypes to invite collaboration and increase diagnostic yield. Another example (also accessible via DECIPHER) is that of Matchmaker Exchange [287] or Genematcher [288], for mining other databases based on matching genotype and phenotype data. VarSome is a data aggregator that encompasses platforms and bioinformatic tools with the aim of consolidating information required for variant analysis. VarSome draws information from 30 databases [289] and is also embedded into another web-based tool, VariantValidator, which enables validation, mapping and formatting of sequence variants using HGVS nomenclature [290].

Knowledge of age-dependent and/or reduced penetrance is vital in reverse-phenotyping as new clinical features may emerge in an individual over time. This is applicable to relatives of PAH patients found to carry presumed pathogenic variants in known disease-associated genes. A detailed evaluation of multiple family members alongside longitudinal follow-up of these individuals is vital to study the disease trajectory which has often higher definition than snapshot phenotyping at diagnosis.

Apart from mining existing databases and health records, recall by genotype (RbG) studies will be a relevant and valuable source of information regarding the biological mechanics of genotype–phenotype associations. By making use of the random allocation of alleles at conception (mendelian randomization—MR) these studies enhance the ability to draw causal inferences in population-based studies and minimise the problems related to confounders; additionally, the focus on phenotypic assessment of a specific carrier subgroup can enhance the understanding of pathobiology in a cost-effective manner. RbG studies can focus on a single variant or multiple variants. The former considers rare and large effect loci to understand the biological pathways of interest, and as a result, the groups are relatively small so the phenotype can be studied in detail, in the latter, genetic variation is used as a proxy of relevant exposure (proviso there is a credible association between genetic markers and exposure). To increase power for studies in which a single variant has a small effect size deployment of aggregate genetic risk scores can be beneficial [291].

### 6.4. Studies

The reversal of the usual hypothesis-driven paradigm for the refined diagnosis was first used by Grunewald and Eklund [278] who by deep clinical phenotyping of patients with Lofgren’s syndrome, showed that erythema nodosum predominantly occurs in women and bilateral periarticular arthritis in men, and that the acute sarcoidosis subgroup can be further characterized according to the presence of HLA-DRB1*0301/DQB1*0201. The reverse phenotyping method has been subsequently adopted in other disease domains (Table 4) and has led to new disease discovery, such as the Koolen–DeVries syndrome, due to a 17q21.31 microdeletion involving the *KANSL1* gene [292], and the Potocki–Lupski Syndrome, due to a 17p11.2 microduplication [293,294]. Recently, a retrospective study of 111 patients with nephrotic syndrome, who underwent WES, showed that reverse phenotyping increased the diagnostic accuracy in patients referred with the diagnosis of steroid-resistant nephrotic syndrome [279].

The reverse phenotyping of *BMPR2* mutation carriers has been studied in PH patients. A large individual participant data meta-analysis found that patients with *BMPR2* mutations have earlier disease onset, worse haemodynamics, are less likely to respond to nitric oxide challenge and have lower survival when compared to those without *BMPR2* mutations [180]. The histological analysis of lungs explanted from those patients also revealed a higher degree of bronchial artery hypertrophy/dilatation, which correlated with the frequency of haemoptysis at presentation [295]. Patients with missense mutations in *BMPR2* that escape nonsense-mediated decay have more severe disease than those with truncating mutations, suggestive of a dominant-negative impact of mutated protein on downstream signalling [296]. However, missense variants in the cytoplasmic tail appear to confer less severe phenotype than other *BMPR2* variants with a later age of onset, milder haemodynamics, and more vasoreactivity [297].

Other good candidates for reverse phenotyping are patients with a mutation in transcription factors, i.e., *TBX4*, which by virtue of gene function would be associated with a more complex and heterogeneous phenotype. Patients with mutations in *TBX4* present with severe PAH associated with bronchial and parenchymal changes, low DLCO, with or without skeletal abnormalities [142], and bimodal age of onset [25]. Interestingly, the penetrance of *TBX4* mutations for skeletal abnormalities is much higher than for PAH [108]. Descriptions of paediatric subjects with mutations in *TBX4,* and likely loci nearby including *TBX2,* characterised by skeletal dysplasias, developmental delay and hearing loss have been reported in CHD and PH patients [298,299]. Interestingly, the US PAH Biobank study confirmed a causative role of *TBX4* not only across various age groups (12/266; 4.6%—paediatric-onset; 11/2345; 0.47%—adult-onset) but also across different PH phenotypes: 2 (one adult; one paediatric)/23; 8.7% PAH-CTD, 3 (two paediatric and one adult-onset)/23; 13% PAH-CHD [25]. Not all patients harbouring deleterious variants in *TBX4* fit neatly into Group 1 PH, some individuals with heterozygous mutations in *TBX4* or large deletions encompassing *TBX4* were characterised by death in infancy secondary to acinar dysplasia [300], congenital alveolar dysplasia and pulmonary hypoplasia [301]. The studies dissecting phenotype by mutation type or looking for subtle features (i.e., SPS) among patients harbouring deleterious variants in *TBX4* who were initially diagnosed with PAH are lacking.

Reverse phenotyping of patients with PTVs in *KDR* revealed mild interstitial lung disease in all index cases and one affected relative [30], which was further confirmed in an independent case report [117].

### 6.5. Limitations, Challenges and Future Directions

Reverse phenotyping has the potential to shed light on genotype–phenotype associations and accelerate the recruitment of homogenous, well-defined groups to clinical trials, thereby increasing the chances of effective treatment and decreasing the risk of side effects. Its success depends on the quality of forward phenotyping, and forward and reverse genetic studies accurately reporting the phenotype of interest (use of standardised vocabulary to describe phenotypic abnormalities (i.e., HPO terms), clinical measurements (i.e., Logical Observation Identifiers (LOINC) [303]) and systematic approaches to data sharing.

In the future, national and international efforts should be directed at optimising recruitment of large homogeneous cohorts for forward genetics studies and, simultaneously, preempting any potential ethical issues related to subsequent recall by genotype studies. Similarly, recall by genotype studies will face challenges related to specific study design, i.e., when recruiting newborns or children who might not express the full phenotype by the time of diagnosis/inclusion.

## 7. Missing Heritability in the Postgenomic Era

As already alluded to in the introduction, missing heritability is the proportion of phenotypic variance not explained by the additive effects of known variants. There are multiple sources of missing heritability beyond the rare and common sequence variation described above (Figure 2). In the following paragraphs, we will discuss studies which contribute to a better understanding of the genetic landscape in PAH, beyond germline, rare, and common sequence variation.

### 7.1. Unknown Genetic Variation

Recently, large case-control studies have allowed us to discover new genes involved in the pathogenesis of PAH; all these studies, however, assumed that PAH is a monogenic condition and examined only protein-coding space. Although this permitted the discovery of more than 16 genes associated with PAH (Table 3), and increased the diagnostic yield, most PAH cases remain unexplained, suggesting additional variation in non-coding regions. It is now known that the remaining 98% of non-coding space plays a crucial role in the regulation of gene expression and requires interpretation using its 3D structure that allows regulatory elements to function across long distances [304]. Regulatory elements such as enhancers, promoters and actively transcribed genes are located in open-chromatin regions so they can be accessed by transcriptional machinery. A number of methods for the identification of non-coding regulatory elements (NCRE) have been developed over the last 20 years [305]. A recent study investigated variation in non-coding regions by researching differences in chromatin marks and gene expression in PAECs from patients with I/HPAH and controls. Using ChiP-Seq profiling of the active histone marks (H3K27ac, H3K4me1, H3K4me3) and RNA-Seq, it was shown that while PAECs from PAH patients and healthy controls are similar in terms of gene expression, they undergo large-scale remodelling of the active chromatin regions, which leads to differential gene expression in response to external stimuli. Overall, this study not only validated and expanded our knowledge regarding the genes involved in the pathogenesis of PAH, but also highlighted that steady-state expression analyses are of limited value in systems where the mechanism involves an aberrant response to stimuli [306].

So far, mostly germline mutations have been implicated in the pathogenesis of PAH but the ever-increasing age of onset may suggest a role of acquired somatic mutations in disease development. While rare somatic mutations of the *BMPR2* gene within pulmonary vasculature cells could cause or aggravate *BMPR2* haploinsufficiency, previous work has failed to find this to be true among those with a germline *BMPR2* mutation [307]. Conversely, a study by Aldred et al. [308] showed a high frequency of genetically abnormal subclones of PAEC from patients with PAH including one patient harbouring a deleterious *BMPR2* variant. Similarly, Drake et al. [309] demonstrated that a somatic deletion on chromosome 13, encompassing *SMAD9*, in PAECs from a patient with PAH associated with CHD could have contributed to the pathogenesis of PAH, and potentially also to CHD [309]. Likewise, postzygotic mutations have been implicated in the pathogenesis of HHT [310] and HHT with concomitant PAH [311,312]. Somatic mutations in *TET2* have recently been reported in PAH cases, and the phenotype was replicated in the mouse model [55].

Impaired bioenergetics is an established feature of PAH [313,314] whether its origins can be traced back to mitochondrial DNA (mtDNA) remains an open question, but a recent study revealed that mitochondrial haplogroups influence the risk of PAH and that susceptibility to PAH emerged as a result of selective enrichment of specific haplogroups upon the migration of populations out of Africa [315]. Studies into the role of mtDNA in PAH will require novel bioinformatics tools and variant prioritisation approaches to account for differences between nuclear and mitochondrial genomes [316].

### 7.2. Epigenetic Inheritance

Epigenetics are heritable changes in gene function that occur without a change in the sequence of DNA and include DNA methylation, histone modification and RNA interference. All three of these modifications have been shown to be involved in the development of PAH. Epigenetic changes can be heritable [317,318], de novo and may be modified by environmental factors such as diet, smoking, and drugs.

#### 7.2.1. DNA Methylation

DNA methylation is a well-described form of epigenetic modification which involves the transfer of a methyl group to cytosine residues in dinucleotide CpG sequences of DNA resulting in gene silencing. In PAH, the best described is hypermethylation of the gene encoding superoxide dismutase 2 (*SOD2*), an enzyme involved in H_2_O_2_ regulation which also acts as a tumour suppressor gene. Hypermethylation of *SOD2* leads to uncontrolled cell proliferation in multiple cancers; similarly reduced expression of *SOD2* has been shown in the rat model of PH [319] and in IPAH [320]. Further evidence comes from a recent study which found rare germline and somatic mutations in the key regulator of DNA demethylation, *TET2*, in 0.39% of PAH cases [55]. The phenotype was replicated in the *Tet^-/-^* mouse and reversed with IL-1β blockade.

#### 7.2.2. Histone Acetylation

Histone acetylation is the most common form of histone modification [321], with a proven impact on PAH development. In rat and human PH histone deacetylases, HDAC1 and HDAC5 concentrations are elevated and histone deacetylase inhibitor can reverse hyperproliferation of PASMCs and decrease mPAP and RV hypertrophy [322].

#### 7.2.3. RNA Interference

A large number of non-coding RNAs (ncRNAs), including long non-coding RNAs (lncRNAs), miRNAs, circular RNAs (circRNAs), and piwi-interacting RNAs (piRNAs) are involved in the regulation of a variety of cellular processes. For example, so far, almost 27,000 lncRNAs have been identified in the human genome [323] and reported regulating gene expression via diverse mechanisms [324]. In PAH, both miRNA and lncRNAs have been studied and reported to impact cells relevant to PAH pathogenesis, including PASMCs, PAECs and fibroblasts. Regulatory effects of lncRNA in PAH range from enhanced proliferation of PASMCs (H19 [325], PAXIP-AS1 [326], MALAT1 [327], Inc-Ang362 [328], Tug1 [329], HOXA-AS3 [330], MEG3 [331], TCONS_00034812 [148], and UCA1 [332]), over suppressed proliferation and/or migration of PASMCs (MEG3 [333,334], CASC2 [335], and LnPRT [336]) to induction of EndoMT in PAECs (MALT1 [337] and GATA6-AS [338]). Significant causative links exist between multiple miRNAs and BMPR2 expression and signalling [339,340,341], as well as hypoxia [342]. Importantly, although attractive as a therapeutic target, miRNAs have not been shown to have the potential to reverse the disease, which is most likely due to the high degree of redundancy and “fine-tuning” rather than “switch on/off” mode of action of these molecules.

Although still limited, our knowledge of ncRNA in the pathobiology of cardiovascular diseases [343,344] including PAH, is rapidly growing. The major limitation of studies so far is that the focus has been mostly on identifying the role of preselected ncRNA with very limited effort to investigate ncRNA in a global manner accounting for multiple interactions.

### 7.3. Interactions

Although difficult to estimate, gene and environment covariation and gene-environment interactions may play a significant role in heritability, expressivity and penetration of the disease. The most commonly studied factors are drugs, diet, toxins, radiation and stress, and some of these factors have been implicated in the development of PAH. Definitive associations between PAH and drugs and toxins based on outbreaks, epidemiological case-control studies or large multicentre series have been confirmed for aminorexigenes, methamphetamines, dasatinib and toxic rapeseed oil, yet some others require further validation [18]. Interestingly, several studies reported that BMPR2 expression and degradation can be affected by viral proteins and cocaine [345,346,347]. Similarly, a number of volatile organic compounds (VOCs) identified in exhaled breath condensate in association with PAH, are exogenous [348] and may be considered pollutants due to exposure to cigarette smoke, air pollution and radiation [349]. Contribution of volatile compounds was also reported in the pathogenesis of PVOD [350].

A specific example of the environmental effects on phenotype is the foetal origins of adult diseases (FOAD) hypothesis. This hypothesis was based on the studies which reported that intrauterine exposure increased the risk of specific cardiovascular and metabolic diseases [351,352,353] as well as impacted on lung function [354,355,356] in adult life. In the area of PH, prenatal exposure to antidepressants and persistent pulmonary hypertension of the newborn (PPHN) has been a subject of multiple studies. A meta-analysis revealed that the risk for PPHN in infants exposed to SSRIs during late pregnancy is small, although significantly increased [357]. In systemic hypertension, the transmission of gut microbiota from parent to offspring was shown to influence the disease risk in adulthood [358] and similarly, bacterial translocation may contribute to PAH development [359]. Air pollution was also shown to correlate with PAH severity and outcomes [360].

With the advent of NGS, the notion of digenic or oligogenic inheritance has gained traction. HPAH has been historically considered a monogenic condition but incomplete penetrance may indicate that other germline or somatic variants are required for the disease to develop. Therefore, at least in a proportion of cases, the inheritance may be digenic or even oligogenic. In the true digenic model, both genes are required to develop the disease. Conversely, in the composite class model, a variant in one gene is sufficient to produce the phenotype, but an additional variant in a second gene impacts the disease phenotype or alters the age of onset [361]. The latter model seems to be plausible in PAH, where co-occurrence of the variants in different PAH risk genes has been reported to impact on disease onset and penetrance [362]. Patients harbouring deleterious variants in more than one PAH risk gene have been reported in case reports [162], small [363] and large cohorts of HPAH patients [24,26,113]. Finally, the phenomenon of synergistic heterozygosity, whereby the effect of susceptibility genes is enhanced by modifier genes, in which common variants may influence the disease onset and severity, has been described in PAH [210].

### 7.4. Population Dependent Heterogeneity

It is well recognised that although rare diseases can occur in any population, some ethnic groups are characterised by their higher incidence, i.e., sickle cell disease is more common in African, African-Americans and Mediterranean populations and similarly, Tay-Sachs disease occurs mostly in people of Ashkenazi Jewish or French Canadian ancestry. National and international PAH registries have shown differences in demographic characteristics between European and East Asian patients. East Asian IPAH cohorts resemble those of European cohorts from 40 years ago; additionally, they show more pronounced female predominance, a younger age of onset and lower comorbidity burden, and importantly, they demonstrate that significantly more cases can be explained by variation in known PAH risk genes (24% [24,25] vs. 39% [26]).

## 8. Summary

In this review, we have summarised how four theoretical and methodological approaches impact genetic discoveries in rare diseases, with a particular focus on PAH. These steps include: (1) forward phenotyping, which refers to clustering patients into homogenous groups likely to share genetic architecture, (2) forward genetics, which aims to identify the candidate genes responsible for the phenotype, (3) reverse genetics, which consists of experimental in vivo and in vitro targeted modifications to candidate genes in order to analyse their phenotypic impact, and (4) reverse phenotyping, which uses genetic marker data to refine phenotype definitions. Whilst these approaches have been employed in PAH research, it is important to mention that at present, the majority of patients diagnosed with PAH do not have a genetic diagnosis, a finding that indicates “missing heritability”. To date, some progress has been made in addressing this issue; however, there is still a way to go and only through the use of large-scale international cohorts will studies have the power to detect novel genetic risk loci underlying PAH pathobiology and to elucidate this missing heritability.

## Figures and Tables

**Figure 1 genes-11-01408-f001:**
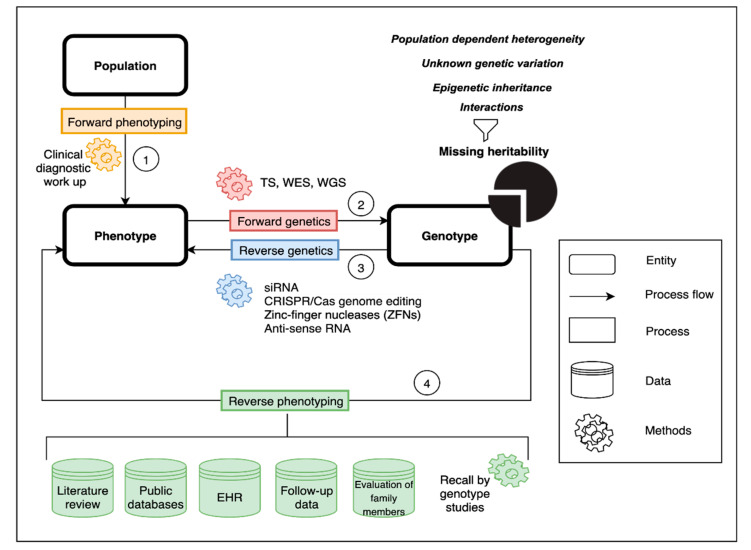
Graphical abstract. Schematic representation of the concepts of forward phenotyping (1) and genetics (2) and reverse genetics (3) and phenotyping (4). Public databases include, but are not limited to, Online Mendelian Inheritance in Man (OMIM), Human Phenotype Ontology (HPO), DatabasE of genomiC varIation and Phenotype in Humans using Ensembl Resources (DECIPHER), ClinVar databases. Forward genetics—‘genotype to phenotype’ approach; reverse genetics—analysis of the impact of induced variation within a specific gene on gene function; reverse phenotyping—clinical assessment directed by genetic results. Abbreviations: TS—targeted sequencing; WES—whole-exome sequencing; WGS—whole-genome sequencing; siRNA—small interfering RNA; EHR—Electronic Healthcare Records.

**Figure 2 genes-11-01408-f002:**
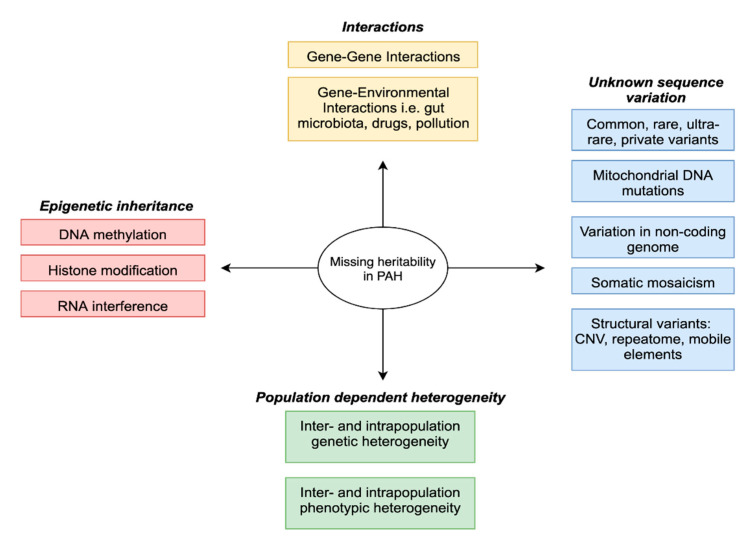
Potential factors contributing to missing heritability in PAH.

**Figure 3 genes-11-01408-f003:**
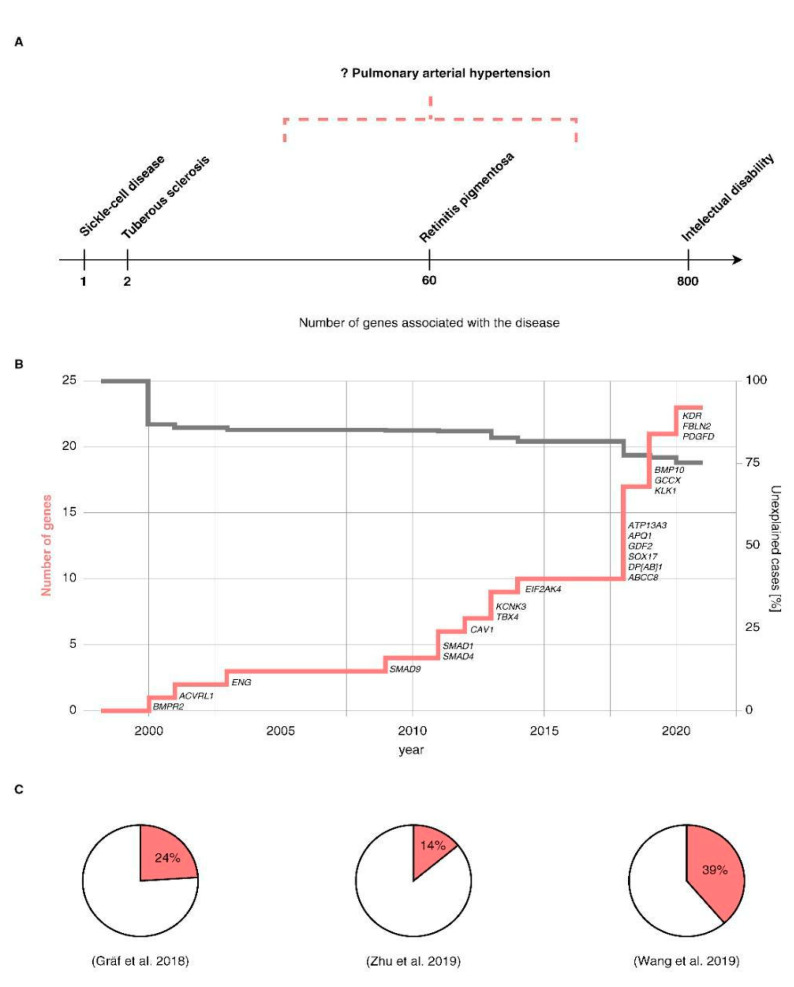
(**A**). Genetic heterogeneity in various genetic disorders, (**B**). Genetic discoveries in PAH, (**C**). Proportion of explained cases by cohort (Gräf et al., 2018—I/HPAH [24]; Zhu et al., 2019—Group 1 PAH [25]; Wang et al., 2019—IPAH [26]). ‘?’ denotes uncertainty around number of genes involved in the pathogenesis of PAH.

**Figure 4 genes-11-01408-f004:**
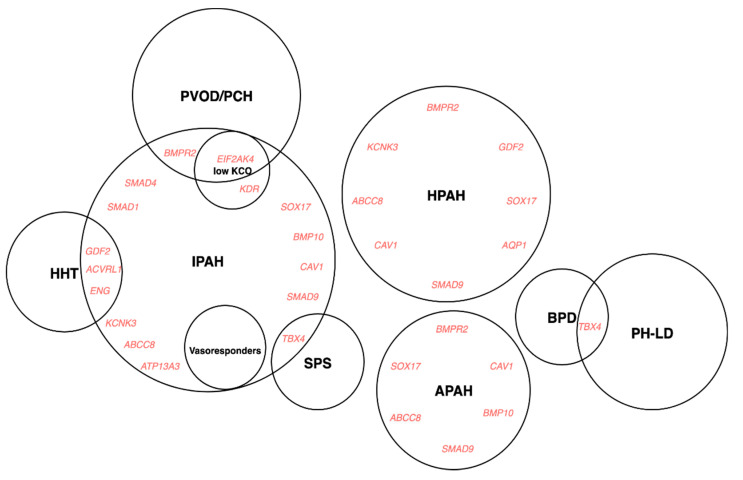
Genotype-phenotype associations in PAH. Abbreviations: I/H/APAH—idiopathic/hereditary/associated pulmonary arterial hypertension; PVOD/PCH—pulmonary veno-occlusive disease/pulmonary capillary haemangiomatosis; SPS—small patella syndrome; BPD—bronchopulmonary dysplasia; HHT—hereditary hemorrhagic telangiectasia.

**Table 1 genes-11-01408-t001:** Definitions and changes to the classification of Group 1 PAH. Abbreviations: WSPH—World Symposium on Pulmonary Hypertension; ERS/ESC—European Respiratory Society and European Society of Cardiology; mPAP—mean pulmonary artery pressure; (R)—resting; (E)—exercise; PAWP—pulmonary artery wedge pressure; PVR—pulmonary vascular resistance; LV—left ventricle; PPH—primary pulmonary hypertension; *BMPR2*—Bone morphogenetic protein receptor, type II; *ACVRL1*—Activin A receptor like type 1; *SMAD9*—SMAD Family Member 9; *CAV1*—Caveolin 1; *KCNK3*—Potassium two pore domain channel subfamily K member 3; PVOD/PCH—pulmonary veno-occlusive disease/pulmonary capillary haemangiomatosis; PAH—pulmonary arterial hypertension; PH—pulmonary hypertension; PPHN—persistent pulmonary hypertension of the newborn; RHC—right heart catheterization.

WSPH Proceedings and ERS/ESC Guidelines	Definition of Group 1	Comments	Changes to the Classification
1st WSPH, Geneva, 1973 [12]	No haemodynamic definition mentioned		
2nd WSPH, Evian, 1998 [13]	No haemodynamic definition mentioned, but RHC recommended for diagnosis		Introduction of the terms primary (PPH) and secondary (related to other conditions) pulmonary hypertension, recognition of familial forms of PH
3rd WSPH, Venice, 2003 [14]	mPAP(R) > 25 mmHg; mPAP(E) > 30 mmHg; PAWP < 15 mmHg; PVR > 3 WU		Abandonment of the term primary pulmonary hypertension, the introduction of terms idiopathic and familial PAH as well as associated PAH, *BMPR2* and *ACVRL1* implicated in the pathogenesis of PAH
4th WSPH, Dana Point, 2008 [5]	mPAP(R) ≥ 25 mmHg; PAWP ≤ 15 mmHg	Exercise-induced PH removed from the definition as although (R) mPAP has been shown to be stable across age groups, (E) mPAP increases with age hence based on the available data it was not possible to define a cutoff	Introduction of the terms idiopathic (no family history, no precipitating risk factor) and hereditary (encompassing familial cases with or without identified germline mutations and PAH). Inclusion of PH associated with Schistosomiasis and PH associated with chronic hemolytic anaemia to Group 1
ERS/ESC Guidelines, 2009 [15]	mPAP(R) ≥ 25 mmHg; PAWP ≤ 15 mmHg; CO normal or reduced	No definition of PH on exercise	
5th WSPH, Nice, 2013 [16]	mPAP(R) ≥ 25 mmHg; PAWP ≤ 15 mmHg; PVR > 3 WU	Introduction of PVR to the definition, a recommendation to report PVR in WU; fluid challenge may be helpful to unmask occult LV diastolic dysfunction	*SMAD9*, *CAV1* and *KCNK3* included as risk genes for HPAH
ERS/ESC Guidelines, 2015 [17]	mPAP(R) ≥ 25 mmHg; PAWP ≤ 15 mmHg	The clinical significance of a mPAP between 21 and 24 mmHg is unclear	Group 1′ PVOD/PCH has been expanded and includes idiopathic, heritable, drug-, toxin- and radiation-induced and associated forms; PPHN includes a heterogeneous group of conditions that may differ from classical PAH. As a consequence, PPHN has been sub-categorised as group I′′.
6th WSPH, Nice, 2018 [18]	mPAP(R) ≥ 20 mmHg; PAWP ≤ 15 mmHg; PVR ≥ 3 WU	PVR ≥ 3WU should be used as a diagnostic criterion for all forms of PH	PAH long-term responses to calcium channel blockers established as a subtype of Group 1; PAH with overt features of venous/capillaries (PVOD/PCH) involvement established as a subtype of Group 1

**Table 2 genes-11-01408-t002:** Landmark forward genetics studies in Group 1 PAH. Abbreviations: *BMPR2*—Bone morphogenic protein receptor type 2; *ENG*—Endoglin; *ACVRL1*—Activin A Receptor Like Type 1; *SMAD*—SMAD Family Member; *CAV1*—Caveolin 1; *KCNK3*—Potassium Two Pore Domain Channel Subfamily K Member 3; *TBX4*—T-Box Transcription Factor 4; *EIF2AK4*—Eukaryotic Translation Initiation Factor 2 α Kinase 4; *GDF2*—Growth Differentiation Factor 2; *SOX17*—SRY-Box Transcription Factor 17; *ATP13A3*—ATPase 13A3; *AQP1*—Aquaporin 1; *ABCC8*—ATP Binding Cassette Subfamily C Member 8; *BMP10*—Bone Morphogenetic Protein 10; *KLK1*—Kallikrein 1; *GCCX*—γ-Glutamyl Carboxylase; *KDR*—Kinase insert domain receptor; *TET2*—Tet Methylcytosine Dioxygenase 2; *FBLN2*—Fibulin 2; *PDGFD*—Platelet-Derived Growth Factor D.

Study (Reference)	Genes	Study Design	Sample	Ethnicity	Method	Reference Genome
Lane et al. 2000, [101]	*BMPR2*	Case-level data	Cases: *n* = 8 PPH kindreds for candidate gene mutational analysis	Not stated	TS	H.sapiens mRNA for BMPR-II: Genbank Z48923
Thomson 2000, [102]	*BMPR2*	Case-level data	Cases: *n* = 50 PPH	Not stated	TS	Not stated
Trembath et al. 2001, [103]	*ACVRL1*	Case-level data	Cases: 5 kindreds plus 1 individual patient with HHT, including *n* = 10 cases with PH	Not stated	TS	Not stated
Chaouat 2004, [104]	*ENG*	Case-level data	Case: *n*=1 HHT, PPH with history of anorexigen use	Not stated	TS	Not stated
Harrison et al. 2005, [105]	*ACVRL1, ENG*	Case-level data	Cases: *n* = 18 I/APAH	Not stated	TS	Not stated
Shintani et al. 2009, [106]	*SMAD9 (SMAD8)*	Case-level data	Cases: *n* = 23 IPAH	Japanese	TS	Not stated
Nasim et al. 2011, [54]	*SMAD1, SMAD4, SMAD9*	Case-level data	Cases: *n* = 324 IPAH/APAH/CTEPH; Controls: *n* = 1584	European & Japanese	TS	Not stated
Austin et al. 2012, [74]	*CAV1*	Case-level data	Cases: 3-generation family, 6 with PAH; Additional cohort: *n* = 260 unrelated I/HPAH cases; Controls: *n* = 1000	European	WES	GRCh37
Ma et al. 2013, [107]	*KCNK3*	Case-level data	Cases: Family in which multiple members had PAH	Not stated	WES	GRCh37
Kerstjens-Frederikse et al. 2013, [108]	*TBX4*	Case-level data	Cases: *n* = 20 childhood-onset I/HPAH; *n* = 49 adult-onset I/HPAH; *n* = 23 SPS	Not stated	TS	Not stated
Eyries et al. 2014, [109]	*EIF2AK4*	Case-level data	Cases: *n* = 13 PVOD families	Not stated	WES	GRCh37
Best et al. 2017, [110]	*EIF2AK4*	Case-level data	Cases: *n* = 81 I/HPAH	Not stated	TS	Not stated
Hadinnapola et al. 2017, [111]	*EIF2AK4*	Case-control data	Cases: *n* = 880 I/FPAH, PVOD/PCH; Controls: *n* = 7134 non-PAH controls and their relatives recruited to NBR	European: 84.6%	WGS	GRCh37
Gräf et al. 2018, [24]	*GDF2, SOX17, ATP13A3, AQP1*	Case-control data	Cases: *n* = 1048 I/F/PAH, PVOD/PCH; Controls: *n* = 7979 non-PAH controls and their relatives recruited to NBR	European: 84.6%	WGS	GRCh37
Zhu et al. 2018, [75]	*SOX17*	Case-control data	Cases: *n* = 256 I/FPAH-CHD; Additional cohort: *n* = 413 I/FPAH screened for rare variants in SOX17; Controls: *n* = 7509 gnomAD	Not stated	WES	GRCh37
Hiraide et al. 2018, [112]	*SOX17*	Case-level data	Cases: *n* = 12 IPAH and 12 family members; Additional cohort: *n* = 128 I/HPAH screened for *SOX17* mutations	Japanese: 100%	WES	Not stated
Bohnen et al. 2018, [113]	*ABCC8*	Case-control data	Cases: *n* = 913; Controls: *n* = 33,369 European adults from ExAC & *n* = 49,630 Europeans from the Regeneron-Geisinger DiscovEHR study	Not stated	WES, WGS	GRCh37
Wang et al. 2019, [26]	*GDF2*	Case-control data	Cases: *n* = 331 IPAH; Controls: *n* = 10,508 from available reference data sets	East Asian: 100%	WES, WGS	GRCh37
Eyries et al. 2019, [114]	*BMP10*	Case-level data	Cases: *n* = 268 I/HPAH, PVOD/PCH	European: > 90%	TS	GRCh37
Hodgson et al. 2020, [115]	*BMP10*	Case-level data	Cases: *n* = 1048 I/FPAH, PVOD/PCH	European: 84.6%	WGS	GRCh37
Zhu et al. 2019, [25]	*KLK1*, *GCCX*	Case-control data	Cases: *n* = 2572 Group 1 PAH; Controls: *n* = 12,771	European: 72%	WES	GRCh38
Rhodes et al. 2019, [116]	*HLA-DPA1/DPB1, SOX17 enhancer*	Case-control data	Cases: *n* = 2085 cases; Controls: *n* = 9659	European: 100%	WGS	GRCh37
Swietlik et al. 2019, [30]	*KDR*	Case-control data	Cases: *n* = 1122 PAH; Controls: *n* = 11,889 non-PAH NBR	European: 84%	WGS	GRCh37
Eyries et al. 2020, [117]	*KDR*	Case-level data	Cases: *n* = 311 unrelated PAH	Not stated	TS	Not stated
Potus et al. 2020, [55]	*TET2*	Case-control data	Cases: *n* = 2572; Controls: *n* = 7509 non-Finnish European subjects from gnomAD	European: 72%	WES	GRCh38
Zhu et al. 2020, [118]	*FBLN2, PDGFD*	Case-control data	Cases: *n* = 4175; Controls: *n* = 18,819 from SPARK and gnomAD cohorts	European: 54.5%	WES	GRCh38

**Table 3 genes-11-01408-t003:** Supporting evidence for the role of risk genes in PAH pathogenesis. Abbreviations: (+) indicates that the paper provides information in favour of the role of the given gene in the pathogenesis of PAH, (−) indicates that the paper does not provide support for the role of the given gene in the pathogenesis of PAH.

	Forward Genetics			Reverse Genetics		
Gene	Case-Control Data	Case-Level Data	Segregation Data	Functional Aberration	Disease Model	Rescue
**MOI: Autosomal Dominant**						
*BMPR2*	(+) [24,25]	(+) [101,114]	(+) [101,122]	(+) [123,124,125,126,127,128,129,130]	Animal: (+) [123,126,128,129,130,131] Cell culture: (+) [124,129,130]	(+) [128,129,130,131] (+)
*ACVRL1*	(+) [24]	(+) [25,103,105,114]	(+) [103]			
*ENG*	(−) [24,25]	(+) [24,75,104]	(+) [104]		Animal: (−) [132]	
*SMAD9*	(−) [24,25]	(+) [24,25,75,106,114]		(+) [54,106,133]	Animal: (+) [134] Cell culture: (+) [54,106,133]	(+) [133,135]
*SMAD1*	(−) [24,25]	(+) [54], (−) [24,25]		(+) [54,136]	Animal: (+) [136] Cell culture: (+) [54]	
*SMAD4*	(−) [24,25]	(+) [25,54], (−) [24]		(+) [54], (−) [54]	Cell culture: (+) [54]	
*CAV1*	(+) [74]	(+) [25,75], (−) [24]		(+) [137]	Animal: (+) [138,139] Cell culture: (+) [137]	(+) [137]
*TBX4*	(+) [24,140]	(+) [75,108,114,141,142]	(+) [143,144]			
*KCNK3*		(+) [24,25,107]	(+) [107]	(+) [107,145]	Animal: (+) [146] Cell culture: (+) [107,145]	(+) [107]
*ATP13A3*	(+) [24]			(+) [147,148]	Animal: (+) [147,148]	
*AQP1*	(+) [24]			(+) [149,150] (−) [149]	Animal: (+) [149,151] Cell culture: (+) [150]	(+) [149]
*GDF2*	(+) [24,25,26]	(+) [114]	(+) [115]	(+) [26,115]	Animal: (−) [152] Cell culture: (+) [26,115]	
*SOX17*	(+) [24,75]	(+) [112]		(+) [153,154]	Animal: (+) [153] Cell culture: (+) [154]	
*ABCC8*	(+) [113]		(+) [113]	(+) [113]	Cell culture: (+) [113]	(+) [113]
*BMP10*		(+) [114,115]				
*GGCX*	(+) [25]					
*KLK1*	(+) [25]					
*KDR*	(+) [30]		(+) [30,117]	(+) [155]	Animal: (+) [155]	
*FBLN2*	(+) [118]					
*PDGFD*	(+) [118]					
*TET2*	(+) [55]			(+) [55]	Animal: (+) [55]	(+) [55]
*BMPR1A*		(+) [25,75]		(+) [156,157,158,159]	Animal: (+) [156,157,159]	(+) [156]
*BMPR1B*		(+) [24,25,75]		(+) [160]	Cell culture: (+) [160]	
*TOPBP1*		(+) [24]				
*THBS1*				(+) [161]	Cell culture: (+) [161]	(+) [161]
*KCNA5*		(+) [162]		(+) [163]	Cell culture: (+) [163]	(+) [163]
**MOI: Autosomal Recessive**						
*EIF2AK4*	(+) [111]	(+) [114]	(+) [109,110]	(+) [109]	Animal: (+) [164]	
**Common Variation**						
*enhancer near SOX17*	(+) [116]		(+) [116]	(+) [116]		
*locus within HLA-DPA1/DPB1*	(+) [116]					
*CBLN2*	(+) [165] (−) [116]					
*SIRT3*		(+) [166]	(+) [166]	(+) [166]	Animal: (+) [166,167]	
*UCP2*				(+) [168]	Animal: (+) [167,169]	
*EDN1*		(+) [170]	(+) [171]			
*AGTR1*	(+) [34]					
*TOPBP1*		(+) [172]		(+) [172]	Cell culture: (+) [172]	
*Endostatin*	(+) [173]					
*TRPC6*				(+) [174]	Cell culture: (+) [174]	

**Table 4 genes-11-01408-t004:** Examples of successful studies that used reverse phenotyping. Abbreviations: *BMPR2*—Bone morphogenic protein receptor type 2; PAH—pulmonary arterial hypertension; WES—whole-exome sequencing, * denotes allele number.

Gene	Condition	Study Design	Data Sources	Results	References
HLA-DRB1*0301/DQB1*0201	Sarcoidosis	Mix of retrospective and prospective cases with acute onset sarcoidosis	Health records	Patients with acute onset sarcoidosis carrying HLA-DRB1*0301/DQB1*0201 genotype have good prognosis, manifestations of Lofgren’s syndrome differ between man and woman.	[278,302]
*BMPR2*	PAH	Retrospective analysis of 169 PAH patients	WES & Health records	Patients with missense mutations that escape nonsense-mediated decay have more severe disease than those with truncating mutations.	[296]
*BMPR2*	PAH	Retrospective analysis of 171 patients		Missense variants in the cytoplasmic tail appear to confer less severe phenotype than other *BMPR2* variants with a later age of onset, milder haemodynamics, and more vasoreactivity.	[297]
*BMPR2*	PAH	A large individual participant data meta-analysis	Literature review	Patients harbouring deleterious *BMPR2* mutations have earlier disease onset, worse haemodynamics, are less likely to respond to NO challenge and have lower survival when compared to those without *BMPR2* mutations.	[180]
*BMPR2*	PAH	Retrospective analysis of 44 PAH patients who underwent lung transplantation between 2005 and 2014	Histology, immunohistochemistry, morphometry of explanted lungs; French registry database	*BMPR2* mutation carriers are more prone to haemoptysis; haemoptysis is closely correlated to bronchial arterial remodelling and angiogenesis; pronounced changes in the systemic vasculature correlate with increased pulmonary venous remodelling, creating a distinctive profile in PAH patients harbouring a *BMPR2* mutation.	[295]
multiple genes	Nephrotic syndrome	A retrospective analysis of all patients diagnosed with nephrotic syndrome between 2000 and 2018	WES & personalised diagnostic workflow	Reverse phenotyping after WES increased the diagnostic accuracy in patients referred with the diagnosis of steroid-resistant nephrotic syndrome.	[279]

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
