# Peer review of "‘There and Back Again’—Forward Genetics and Reverse Phenotyping in Pulmonary Arterial Hypertension"

_genes, 2020, doi:10.3390/genes11121408_

Round 1

Reviewer 1 Report

In their very comprehensive review article, Swietlik and colleagues discuss current knowledge and potential future strategies to further elucidate the genetic architecture underlying pulmonary vascular diseases (PVD), particularly pulmonary arterial hypertension. The authors cover every important aspect related to genetic discovery in PVD and discuss four strategies to close our heritability knowledge gap in PVD/PAH pathobiology. 

The authors are to be commended for crafting such an extensive and detailed yet easily comprehensible article. While I have really enjoyed the read there are a few minor issues that should be addressed:

  • The authors might want to revise and tighten their section on phenotypic heterogeneity in relation to the history of diagnostic criteria (l. 100 ff). While it is important to mention that the hemodynamic definition of PH contributes to its phenotypic heterogeneity, initial estimates indicate that the increase in incident PH patients due to the revised diagnostic mPAP threshold might be relatively small. The same issue also comes up later in the manuscript (l. 835 ff). Also, I don't see the additional benefit of Table 1 which appears somewhat out of scope.  
  • The authors might want to add a brief definition of cis and trans configuration (l. 257) to make the sentence easier to understand.
  • References appear to be missing for PPARG and KLF4 (l. 799ff).
  • The reverse genetics in vitro section (l. 872 ff) is incomplete yet feels strangely detailed. In addition, the way l. 885ff is currently worded is confusing, as activity of the effector Caspase 3 usually functions as the luminescent readout and not as pro-apoptotic stimulus. In general, I would recommend revising this section heavily or considering taking it out altogether.  
  • The authors might want to clarify that PH is also present in PVOD (l. 748 ff)
  • I have realized that none of the recent seminal works on mutant BMPR2 dysfunction by the Stanford groups of Dr. Nicholls, Dr. Rabinovitch and Dr. Gu have been cited (e.g. Table 3, l. 906 ff, l. 959 ff, l. 984ff, l. 1261 ff, l. 1345ff). The authors might want to consider adding these works if they see fit. 

Author Response

The authors might want to revise and tighten their section on phenotypic heterogeneity in relation to the history of diagnostic criteria (l. 100 ff). While it is important to mention that the hemodynamic definition of PH contributes to its phenotypic heterogeneity, initial estimates indicate that the increase in incident PH patients due to the revised diagnostic mPAP threshold might be relatively small. The same issue also comes up later in the manuscript (l. 835 ff). Also, I don't see the additional benefit of Table 1 which appears somewhat out of scope.

Author's response: Thank you for your comment. This review was targeted at individuals who may be new to the field and/or require a broad overview of PAH. Such a compendium of the different clinical criteria over time is not recorded as comprehensively elsewhere, to our knowledge. We, therefore, believe Table 1 provides this overview in a compact way. Whilst we agree that the increase in incident PH patients due to revised diagnostic thresholds may be relatively small, we feel it is important to acknowledge the change in these definitions over time. To not overemphasise the effect on sample size, however, we have removed this statement from the limitations paragraph (l. 835 ff).

The authors might want to add a brief definition of cis and trans configuration (l. 257) to make the sentence easier to understand.

Author's response: We thank the reviewer for this suggestion and added respective statements.

References appear to be missing for PPARG and KLF4 (l. 799ff).

Author's response: We thank the reviewer for spotting this. While the reference by Villar et al was used a sentence above we acknowledge that this reference was only pointing to a conference abstract. Hence, we have added a couple of addtional references to reinforce the Villar et al finding. 

The reverse genetics in vitro section (l. 872 ff) is incomplete yet feels strangely detailed. In addition, the way l. 885ff is currently worded is confusing, as activity of the effector Caspase 3 usually functions as the luminescent readout and not as pro-apoptotic stimulus. In general, I would recommend revising this section heavily or considering taking it out altogether. 

Author's response: Thank you for pointing out the confusion caused by the sentence regarding Caspase-3. After your comments, we decided to shorten the in vitro paragraph and reference reviews which discuss molecular aspects of PAH.

The authors might want to clarify that PH is also present in PVOD (l. 748 ff)

Author's response: We much appreciate this comment and have moved the last sentence of the paragraph up to the beginning of it in order to clarify the change upfront.

I have realized that none of the recent seminal works on mutant BMPR2 dysfunction by the Stanford groups of Dr. Nicholls, Dr. Rabinovitch and Dr. Gu have been cited (e.g. Table 3, l. 906 ff, l. 959 ff, l. 984ff, l. 1261 ff, l. 1345ff). The authors might want to consider adding these works if they see fit. 

Author's response: Thank you for pointing out that these references were missing. We have now included a rat and cell culture model of disease in Table 3, from the authors mentioned and have referenced them also in the text where appropriate.

Reviewer 2 Report

I congratulate you on your superbly written and encompassing review.

Really, the only critiques I have are length and the need for the GRCh37-38 comparison on page 12.  

With regards to length, I think you could have conveyed a still-complete message in a much smaller footprint.  But, you nailed the science.

Author Response

Really, the only critiques I have are length and the need for the GRCh37-38 comparison on page 12.

Author's response: We appreciate the reviewer's comment and have removed some sections from the manuscript also in response to Reviewer 1. Regarding the GRCh37 vs. GRCh38 comparison we still consider this as an important message providing relevant considerations for the analysis approaches. We have, however, revised and shortened the section on pan-genomes and graph-based representation of genomes to address this suggestion.